# Identifying and Correcting Label Noise for Robust GNNs via Influence Contradiction

**Wei Ju** [1] **Wei Zhang** [2] **Siyu Yi** [*3] **Zhengyang Mao** [4] **Yifan Wang** [5] **Jingyang Yuan** [4]
**Zhiping Xiao** [6] **Ziyue Qiao** [7] **Ming Zhang** [*4]

## Abstract

Graph Neural Networks (GNNs) have shown remarkable capabilities in learning from graph-structured data with various applications such as social analysis and bioinformatics. However, the presence of label noise in real scenarios poses a significant challenge in learning robust GNNs, and their effectiveness can be severely impacted when dealing with noisy labels on graphs, often stemming from annotation errors or inconsistencies. To address this, in this paper we propose a novel approach called ICGNN that harnesses the structure information of the graph to effectively alleviate the challenges posed by noisy labels. Specifically, we first design a novel noise indicator that measures the influence contradiction score (ICS) based on the graph diffusion matrix to quantify the credibility of nodes with clean labels, such that nodes with higher ICS values are more likely to be detected as having noisy labels. Then we leverage the Gaussian mixture model to precisely detect whether the label of a node is noisy or not. Additionally, we develop a soft strategy to combine the predictions from neighboring nodes on the graph to correct the detected noisy labels. At last, pseudo-labeling for abundant unlabeled nodes is incorporated to provide auxiliary supervision signals and guide the model optimization. Experiments on benchmark datasets show the superiority of our approach over competitive baselines in noisy label scenarios. The source code is available at: https://github.com/wayc04/ICGNN.

[1]School of Artificial Intelligence, Sichuan University, China [2]College of Computer Science, Sichuan University, China [3]School of Mathematics, Sichuan University, China [4]National Key Laboratory for Multimedia Information Processing, School of Computer Science, Peking University, China [5]University of International Business and Economics, China [6]University of Washington, USA [7]Great Bay University, China. Correspondence to: Siyu Yi <siyuyi@scu.edu.cn>, Ming Zhang <mzhang_cs@pku.edu.cn>.

*Proceedings of the $43^{rd}$ International Conference on Machine Learning*, Seoul, South Korea. PMLR 306, 2026. Copyright 2026 by the author(s).

## 1. Introduction

Graphs have emerged as a foundational paradigm in machine learning and data mining recently, capturing intricate relationships among entities in diverse domains such as social networks, biology, and recommender systems. Graph Neural Networks (GNNs) have revolutionized the field by enabling effective learning from graph data (Ju et al., 2025), whose key idea is to iteratively update node representations based on the information propagated from neighboring nodes (Gilmer et al., 2017), effectively exploring complex relationships and patterns within graph-structured data.

While GNNs have shown impressive performance in various tasks, they typically hinge on the assumption of clean and accurate class labels. However, real-world graph data often exhibit noisy labels, stemming from reasons such as human errors, inconsistencies in data collection, or subjective interpretations (Song et al., 2022). Consider a recommender system on a user-item interaction graph: noisy labels might arise from incorrect user feedback or mismatches between user preferences and actual behavior. Similarly, in a molecular graph, errors in chemical annotation could lead to misclassification of compounds. Such noisy labels can severely undermine the robustness of GNNs (Yuan et al., 2023a). Moreover, obtaining ground-truth labels for graphs can be expensive and labor-intensive (Luo et al., 2023c), particularly when dealing with graphs with complex topological structures (Ju et al., 2024b). Thus, a robust algorithm that effectively handles noisy labels and label scarcity is crucial to fully unlock the potential of GNNs in real-worlds.

Actually, there are a variety of strategies within the domain of computer vision to mitigate the effects of noisy labels effectively, which can be categorized into three groups: *sample selection*, *loss correction* and *label correction*. The sample selection strategy (Han et al., 2018; Yu et al., 2019; Liang et al., 2024; Pan et al., 2025) aims to filter out noisy samples during training. The loss correction strategy (Wang et al., 2019; Wilton & Ye, 2024; Nagaraj et al., 2025) modifies the loss function to penalize the influence of noisy labels. And label correction techniques (Sheng et al., 2017; Song et al., 2019; Li et al., 2024a) attempt to directly modify noisy labels to improve accuracy.

However, these algorithms often struggle to adapt seamlessly to the graph-structured data due to the complex topological structures, and only a handful of approaches have been developed to tackle noisy labels on graphs (Dai et al., 2021; Du et al., 2023; Qian et al., 2023; Chen et al., 2024; Ding et al., 2024; Li et al., 2025). For example, NRGNN (Dai et al., 2021) constructs edges between unlabeled nodes and labeled nodes with similar features, thereby facilitating predictive precision and credibility of label information. Building on this, RTGNN (Qian et al., 2023) proposes self-reinforcement and consistency regularization as auxiliary supervision to achieve better robustness. Meanwhile, CGNN (Yuan et al., 2023b) enhances node representation robustness via contrastive learning and effectively detects noisy labels using a homophily-driven sample selection strategy on graphs. Recently, ProCon (Li et al., 2025) identifies mislabeled nodes by measuring label consistency among peers, while DREAM (Zhao et al., 2026) employs a dual-standard selection of anchor nodes to assess semantic homogeneity for dynamic, relation-informed optimization.

Despite their efficacy for handling noisy labels on graphs, there still exist some inherent issues: **(i) they lack an effective mechanism to accurately identify nodes with noisy labels and often fail to explicitly incorporate the characteristics of graph structures.** For instance, NRGNN overlooks the issue of noisy label detection by simply connecting unlabeled nodes with similar labeled nodes. It alleviates the impact of noisy labels, but it does not actively detect or address noise in the labels. In contrast, RTGNN and CGNN employ traditional prediction consistency techniques for detection in a direct manner which fail to leverage the rich structural information inherent in graphs, resulting in suboptimal results; **(ii) these methods lack a robust strategy for correcting noisy labels.** For example, RTGNN and ProCon merely reduces the weights of detected noisy labels, which can mitigate their impact but doesn't directly correct the underlying label noise. On the other hand, CGNN employs a neighbor voting mechanism that depends heavily on class distribution and can suffer from sample imbalance, thus easily leading to confirmation errors (Nickerson, 1998).

To address these issues, in this paper we present a novel graph neural network, referred to as ICGNN, which quantifies the influence contradiction among diverse classes of nodes built upon the intricate graph structure. Specifically, to accurately detect potential noisy labels on the graph, we develop an effective noise indicator that employs the graph diffusion matrix to measure the influence contradiction score (ICS) of a node based on interactions with nodes of different classes at both the structure and attribute levels, thereby assessing the credibility of nodes with clean labels. In this way, nodes with higher ICS values are more likely to be identified as having noisy labels. Afterward, the Gaussian mixture model (GMM, Richardson & Green

(1997)) is adopted to precisely detect the presence of noisy labels for nodes. Moreover, based on the detected noisy labels, we design a soft and thus robust correction strategy that integrates predictions from neighboring nodes in the graph to rectify incorrect labels, thus effectively alleviating the impacts posed by noisy labels. Lastly, we incorporate pseudo-labeling techniques for abundant unlabeled nodes to provide additional supervision and overcome the effects of label scarcity. Experimental results across various benchmark graph datasets showcases the effectiveness of our ICGNN in handling label noise under different noise rates and label rates, and achieving superior performance compared to competitive baseline approaches.

## 2. Problem Definition & Preliminaries

**Notations.** Let $\mathcal{G} = \{\mathcal{V}, \mathbf{A}, \mathbf{X}, \mathbf{Y}_{\mathrm{L}}\}$ denote a graph with $N$ nodes and $C$ classes, where $\mathcal{V} = \mathcal{V}_{\mathrm{L}} \cup \mathcal{V}_{\mathrm{U}} = \{v_1, \ldots, v_N\}$ is the node set containing limited labeled nodes in $\mathcal{V}_{\mathrm{L}} = \{v_1, \ldots, v_L\}$ and abundant unlabeled nodes in $\mathcal{V}_{\mathrm{U}} = \{v_{L+1}, \ldots, v_N\}$ ($N - L \gg L$), and $\mathbf{X} \in \mathbb{R}^{N \times F}$ is the node feature matrix. $\mathbf{A}$ is adjacency matrix where $\mathbf{A}_{ij} = 1$ if there is an edge between nodes $i$ and $j$. $\mathbf{Y}_{\mathrm{L}} = \{\mathbf{y}_1, \ldots, \mathbf{y}_L\} \in \{0, 1\}^C$ is one-hot labels of labeled nodes in $\mathcal{V}_{\mathrm{L}}$, which is disturbed by noise.

**Problem Definition.** Given a graph $\mathcal{G} = \{\mathcal{V}, \mathbf{A}, \mathbf{X}, \mathbf{Y}_{\mathrm{L}}\}$ where the labels $\mathbf{Y}_{\mathrm{L}}$ are contaminated by noise and the labeled nodes in $\mathcal{V}_{\mathrm{L}}$ is limited. This paper studies the problem of node classification in semi-supervised scenarios where the goal is to learn a robust GNN, such that the trained GNN can accurately make predictions on unlabeled nodes in $\mathcal{V}_{\mathrm{U}}$.

**GNN-based Encoder.** The fundamental concept of GNNs (Kipf & Welling, 2017) involves updating node representations by aggregating messages from neighboring nodes using a graph convolution operation through the graph structure, following the message-passing mechanisms (Gilmer et al., 2017). Formally, the node representations $\mathbf{Z} = [\mathbf{z}_1, \ldots, \mathbf{z}_{|\mathcal{V}|}]^\top \in \mathbb{R}^{|\mathcal{V}| \times d}$ can be updated as:

$$\mathbf{Z} = \sigma(\hat{\mathbf{A}} \mathbf{X} \mathbf{W}), \quad \hat{\mathbf{A}} = \tilde{\mathbf{D}}^{-\frac{1}{2}} \tilde{\mathbf{A}} \tilde{\mathbf{D}}^{-\frac{1}{2}}, \tag{1}$$

where $\tilde{\mathbf{A}} = \mathbf{A} + \mathbf{I}$, $\tilde{\mathbf{D}}$ is the degree matrix of $\tilde{\mathbf{A}}$. $d$ denotes the dimension of the hidden node representations, $\mathbf{W}$ is the trainable weight matrix, and $\sigma(\cdot)$ is the activation function.

## 3. Methodology

In this section, we introduce the details of our ICGNN for training a robust GNN capable of handling both noisy and limited labels. Our framework ICGNN is composed of three main components designed to address the aforementioned challenges, i.e., (i) noise detection by influence contradiction; (ii) noise cleaning by neighbor aggregation; (iii) optimization against noisy labels. In the subsequent sections, we elaborate on each of these components in detail.

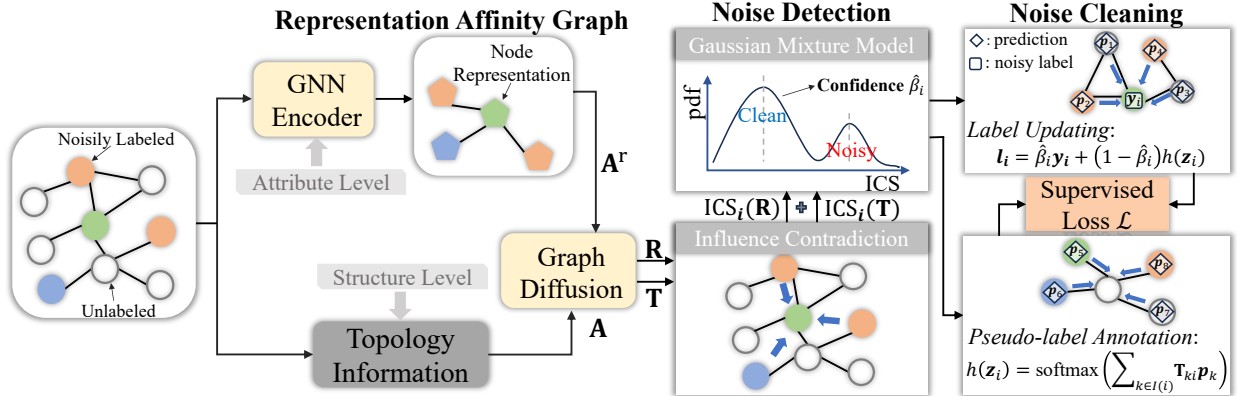

*Figure 1.* Illustration of the proposed framework ICGNN.

## 3.1. Noise Detection by Influence Contradiction

To alleviate the negative effect of the noisy labels, how to effectively detect them is a key factor in the graph with a complex topology. Due to the inherent edge connections, each node in the graph can influence its surrounding neighbors through message passing (Gilmer et al., 2017). Moreover, the homophily assumption in the graph domain states that connected nodes tend to belong to the same class. Hence, based on the above fact and assumption, we reasonably claim that if a labeled node $v \in \mathcal{V}$ encounters strong influence from the nodes belonging to other classes, that is, the node $v$ is subject to a large influence contradiction in the message passing process, then we hold the opinion that the node possibly possesses a noisy label. Based on this hypothesis, we propose a novel noise indicator called *influence contradiction score* to quantify the credibility of nodes with clean labels. The smaller the influence contradiction score, the more likely the node label is to be clean.

Technically, we first leverage the idea of graph diffusion (Klicpera et al., 2019) to globally acquire each node's influence on other nodes based on the graph topology, i.e., the graph diffusion matrix is defined as:

$$\mathbf{T} = \epsilon(\mathbf{I} - (1-\epsilon)\hat{\mathbf{A}})^{-1}, \qquad (2)$$

where the personalized PageRank (Page et al., 1999) is adopted with teleport probability $\epsilon \in (0, 1)$. $\hat{\mathbf{A}}$ is the normalized adjacency matrix in Eq. (1). Each row in the graph diffusion matrix $\mathbf{T}$ can be regarded as the influence distribution exerted outward from each node (Bojchevski et al., 2020; Chen et al., 2021). Grounded in this, we develop the *influence contradiction score* (ICS) for the $i$-th node in $\mathcal{V}_L$:

$$\text{ICS}_i(\mathbf{T}) = \sum_{j=1, j \neq y_i}^{C} \frac{1}{|\mathcal{C}_j|} \sum_{k \in \mathcal{C}_j} \mathbf{T}_{ki}, \qquad (3)$$

where $y_i$ is the noisily annotated label of the $i$-th node, $\mathcal{C}_j$ contains the indices of nodes belonging to the $j$-th class, and $\mathbf{T}_{ki}$ is the $(k, i)$-th entry in matrix $\mathbf{T}$. The normalization term $1/|\mathcal{C}_j|$ is used to eliminate the effect of the number

of nodes belonging to different classes, which makes the ICSs from different classes comparable. In this way, the value of $\text{ICS}_i(\mathbf{T})$ aggregates the influence from the labeled nodes in other classes to the $i$-th node, which granularly characterizes the contradictory influence between the $i$-th node and the nodes annotated by other labels.

However, such a definition in Eq. (3) only relies on the graph structure, while overlooking the effect of the node attribute information. To address this issue, we leverage the node representations to achieve the influence contradiction at the attribute level. Specifically, after feeding given graph $\mathcal{G}$ into a GNN-based encoder (e.g., GCN) to extract the labeled node representations $\mathbf{Z}_L = \{\mathbf{z}_1, \ldots, \mathbf{z}_L\}$, we construct the representation affinity graph based on $\mathbf{Z}_L$, whose adjacent matrix $\mathbf{A}^r \in \{0, 1\}^{L \times L}$ is built by selecting each node representation's $K$ nearest neighbors. We utilize graph diffusion again as Eq. (2) except that $\mathbf{A}$ is replaced by $\mathbf{A}^r$, resulting in the attribute-level diffusion matrix $\mathbf{R}$. Then we define the corresponding ICS for the $i$-th node in $\mathcal{V}_L$:

$$\text{ICS}_i(\mathbf{R}) = \sum_{j=1, j \neq y_i}^{C} \frac{1}{|\mathcal{C}_j|} \sum_{k \in \mathcal{C}_j} \mathbf{R}_{ki}, \qquad (4)$$

where $\mathbf{R}_{ki}$ is the $(k, i)$-th entry in matrix $\mathbf{R}$.

To fuse the structural and attributive information, we combine Eq. (3) and Eq. (4) to derive the final ICS measurement to assist in noise detection, which is formulated as:

$$\text{ICS}_i = (1-\alpha)\text{ICS}_i(\mathbf{T}) + \alpha\text{ICS}_i(\mathbf{R}), \qquad (5)$$

where $\alpha$ is the hyper-parameter to adjust the relative importance of the structure- and attribute-level ICS values, setting 0.5 in experiments. Based on the aforementioned discussion, we can effectively use ICS to determine the credibility of a node's label, whether it is clean or potentially contaminated. In other words, the larger the value of $\text{ICS}_i$, the less confident the label of the $i$-th node is clean. Below we present the theoretical justification of ICS. Here we assume that both $\mathbf{T}$ and $\mathbf{R}$ are column-normalized, i.e., $\sum_{k=1}^{N} \mathbf{T}_{ki} = \sum_{k=1}^{N} \mathbf{R}_{ki} = 1$.

**Theorem 3.1.** *Assume that there exists a constant $\delta \in (0, 1]$ such that for any node, we have*

$$\frac{1}{|\mathcal{C}_{y_i^*}|} \sum_{k \in \mathcal{C}_{y_i^*}} \alpha \mathbf{T}_{ki} + (1 - \alpha)\mathbf{R}_{ki} \geq \delta,$$

*where $y_i^*$ is the true label of the $i$-th node. If $y_i = y_i^*$ (clean), then $ICS_i \leq 1 - \delta$; if $y_i \neq y_i^*$ (noisy), then $ICS_i \geq \delta$.*

The assumption in Theorem 3.1 specifies the requirement on joint structural- and attribute-level homophily, and the proof is shown in Appendix A. Theoretically, when $\delta > 1/2$, the ICS intervals for clean and noisy nodes, $[0, 1 - \delta]$ and $[\delta, 1]$, do not overlap, enabling direct separation by setting a threshold. When $\delta$ is relatively small, the expected values of the two classes still differ, but it becomes challenging to establish a meaningful threshold for effective discrimination.

Further, the value of $\delta$ is unknown in practice. Therefore, to more precisely detect noisy labels, we adopt a GMM to fit the ICS values, where the expectation-maximization (EM) algorithm (Dempster et al., 1977) is employed to achieve the assignment probability (i.e., confidence) that a node has a clean label. EM has the advantage of enabling learnable soft-threshold clustering, avoiding the need for manually-set hard thresholds. Specifically, we consider a two-component GMM, and introduce the latent variables $a_{iq}, i = 1, \ldots, L; q = 1, 2$ to optimize the model using EM, where $a_{iq}$ represents the probability of the $i$-th node being assigned to the $q$-th component. In the E step, we compute the posterior assignment probability by:

$$\beta(a_{iq}) = p(a_{iq} = 1 | \text{ICS}_i, \boldsymbol{\theta}) = \frac{\pi_q \mathcal{N}(\text{ICS}_i | \mu_q, \sigma_q)}{\sum_{q'=1}^{2} \pi_{q'} \mathcal{N}(\text{ICS}_i | \mu_{q'}, \sigma_{q'})};$$

In the M step, we update the mean parameter as:

$$\mu_q = \frac{1}{\sum_{i=1}^{L} \beta(a_{iq})} \sum_{i=1}^{L} \beta(a_{iq}) \text{ICS}_i,$$

where $\mathcal{N}(\cdot | \mu_q, \sigma_q)$ denotes the probability density function of the $q$-th Gaussian, the updates of the assignment parameter $\pi_q$ and variance parameter $\sigma_q$ are omitted for space saving. After several iterations of the E step and M step, the algorithm will converge eventually with theoretical guarantees. We leverage the well-trained posterior assignment probability $\hat{\beta}_i = \hat{\beta}(a_{i\hat{q}})$ as the confidence that the $i$-th labeled node has a clean label, where $\hat{q} = \arg\min_q \hat{\mu}_q$, $\hat{\mu}_q$ is the converged mean parameters for $q = 1, 2$. Such an operation relies on the previous analysis that nodes with smaller ICSs are more likely to have a clean label.

### 3.2. Noise Cleaning by Neighbor Aggregation

Based on the detection results, how to correct these noisy labels is crucial to improve the performance and robustness of the model. Instead of adopting neighbor voting (Yuan et al., 2023b) to compulsively correct them, we consider a softer approach that combines the noisy labels (for labeled nodes) and the neighbors' prediction information. It is a conservative and cautious strategy and thus a robust way, which helps alleviate the confirmation bias problem.

Concretely, at the $t$-th epoch, for the $i$-th ($i \in \{1, \ldots, L\}$) labeled node in $\mathcal{V}_\text{L}$, we consider a convex combination of the one-hot noisy label $\mathbf{y}_i$ and the neighbor prediction information $h^{(t)}(\mathbf{z}_i)$ with $\mathbf{z}_i \in \mathbf{Z}_\text{L}$ as follows,

$$\mathbf{l}_i^{(t)} = \hat{\beta}_i^{(t)} \mathbf{y}_i + (1 - \hat{\beta}_i^{(t)}) h^{(t)}(\mathbf{z}_i), \qquad (6)$$

where we utilize the trained confidence $\hat{\beta}_i^{(t)}$, which represents the credibility of a node having a clean label at the $t$-th epoch, as the weight of keeping the original annotated label in the updated label $\mathbf{l}_i^{(t)}$. As for the expression of $h^{(t)}(\mathbf{z}_i)$, we aggregate the prediction information of the neighbors of the node. Mathematically, denote by $\mathbf{p}_i^{(t)}$ the classifier's softmax-based prediction derived from $\mathbf{z}_i$ at the $t$-th epoch, and then $h^{(t)}(\mathbf{z}_i)$ is defined as:

$$h^{(t)}(\mathbf{z}_i) = \text{softmax}\left(\sum_{k \in I(i)} \mathbf{T}_{ki} \mathbf{p}_k^{(t)}\right), \qquad (7)$$

where $\text{softmax}(\cdot)$ represents the softmax operation, $\mathbf{T}_{ki}$ is the $(k, i)$-th entry in the graph diffusion matrix $\mathbf{T}$ in Eq. (2) to assign weights of other connected nodes, and $I(i)$ is the set of indices sampled from the $i$-th row of $\mathbf{T}$, which has been normalized into a distribution. Such a definition in Eq. (7) encourages the updated label can be corrected by its global neighbors' predictions.

In addition to noise cleaning, due to the limited labels, we also strive to take full advantage of the unlabeled nodes in $\mathcal{V}_\text{U}$ for better model robustness. At the $t$-th epoch, based on the node representations $\mathbf{Z}_\text{U} = \{\mathbf{z}_{L+1}, \ldots, \mathbf{z}_N\}$ from GNN-based encoder, we leverage the same strategy as Eq. (7) to artificially annotate those unlabeled nodes with pseudo labels $h^{(t)}(\mathbf{z}_i), i = L + 1, \ldots, N$, which is capable of providing auxiliary supervision signals to better guide model optimization, which will be discussed in the next section.

### 3.3. Optimization against Noisy Labels

Depending on the proposed noise cleaning for noisily labeled nodes and pseudo-labeling for abundant unlabeled nodes, we utilize the cross-entropy loss to guide the model training, i.e., at the $t$-th epoch, the training is optimized by:

$$\mathcal{L} = \sum_{i=1}^{L} \mathbf{l}_i^{(t)} \log \mathbf{p}_i^{(t)} + \sum_{i=L+1}^{N} h^{(t)}(\mathbf{z}_i) \log \mathbf{p}_i^{(t)}, \quad (8)$$

where $\{\mathbf{p}_i^{(t)}, i = 1, \ldots, N\}$ are the classifier's softmax-based predictions of all nodes in the $t$-th epoch. After converging, we make predictions on the unlabeled nodes in $\mathcal{V}_\text{U}$ based on the node representations $\mathbf{Z}_\text{U}$. Moreover, we follow

*Table 1.* Performance on six datasets (mean±std). The best and runner-up results in all the methods are highlighted with **bold** and underline, respectively. The noise rate is set to 20% as default.

| | Methods | GCN | Forward | Coteaching+ | NRGNN | RTGNN | CGNN | CR-GNN | DND-NET | ProCon | ICGNN |
| | Year | ICLR'17 | CVPR'17 | ICML'19 | KDD'21 | WSDM'23 | ICASSP'23 | NN'24 | KDD'24 | IJCAI'25 | (Ours) |
|---|---|---|---|---|---|---|---|---|---|---|---|
| **Uniform Noise** | Coauthor CS | 80.3±1.4 | 80.5±1.2 | 80.7±1.4 | 83.2±0.5 | 86.7±0.9 | 84.1±0.4 | 82.9±2.3 | 86.2±2.7 | 85.4±1.8 | **87.4±0.8** |
| | Amazon Photo | 82.2±0.9 | 82.1±0.4 | 78.5±0.6 | 83.7±3.9 | 84.8±3.3 | 85.3±0.9 | 81.5±4.6 | 82.3±1.6 | 83.5±2.2 | **87.3±0.5** |
| | Cora | 70.3±1.8 | 73.7±0.7 | 73.6±1.7 | 80.0±0.5 | 79.1±0.5 | 76.8±0.5 | 79.1±4.2 | 76.5±2.0 | 78.6±1.6 | **80.9±0.8** |
| | Pubmed | 77.3±0.9 | 77.4±0.5 | 78.6±0.4 | 79.0±1.6 | 79.8±1.3 | 78.1±0.4 | 80.1±0.9 | 79.4±2.2 | 79.1±1.2 | **80.3±0.5** |
| | DBLP | 71.0±1.5 | 72.4±0.7 | 73.5±1.3 | 79.3±0.8 | 79.0±1.1 | 78.9±0.6 | 79.2±1.3 | 77.0±1.5 | 77.2±1.4 | **80.1±0.6** |
| | Citeseer | 64.9±1.7 | 65.7±2.1 | 66.4±1.3 | 70.1±1.7 | 68.2±3.8 | 69.7±1.3 | 69.3±2.1 | 70.4±3.4 | 68.4±0.9 | **71.5±0.5** |
| **Pair Noise** | Coauthor CS | 79.5±1.1 | 80.5±0.8 | 77.6±3.3 | 83.7±0.9 | 83.8±2.1 | 81.0±1.1 | 81.7±3.1 | 84.0±3.6 | 82.6±1.9 | **85.9±0.8** |
| | Amazon Photo | 80.9±1.2 | 78.7±0.3 | 75.5±1.8 | 83.5±3.6 | 84.2±2.7 | 85.1±0.7 | 78.1±5.2 | 80.1±2.4 | 81.4±2.5 | **86.3±0.6** |
| | Cora | 74.1±0.7 | 76.0±0.7 | 73.8±1.4 | 78.6±0.4 | 77.8±0.7 | 77.5±0.4 | 78.2±3.2 | 75.1±2.7 | 76.3±2.0 | **79.4±0.7** |
| | Pubmed | 78.0±0.4 | 79.6±0.2 | 78.5±0.1 | 79.2±0.7 | 80.4±1.6 | 78.6±0.4 | 80.1±1.2 | 77.8±1.4 | 76.8±1.7 | **80.6±0.3** |
| | DBLP | 72.5±1.2 | 74.4±0.5 | 72.7±1.2 | 79.3±0.9 | 78.4±2.6 | 79.6±0.5 | 78.9±1.7 | 76.5±2.3 | 77.1±1.3 | **80.2±0.4** |
| | Citeseer | 60.3±1.0 | 61.6±0.4 | 65.1±2.1 | 67.8±3.0 | 67.0±2.8 | 66.0±1.7 | 68.6±1.9 | 69.6±1.8 | 67.8±1.1 | **70.7±0.7** |

Dai et al. (2021) and incorporate a reconstruction objective based on negative sampling to better achieve robust classification against noise and with limited labels. The complexity analysis is provided in Appendix C and the optimization process is summarized in Algorithm 1 of Appendix D.

## 4. Experiment

### 4.1. Experimental Setup

**Datasets.** We use six benchmark datasets for evaluation, including one author network: Coauthor CS, one co-purchase network: Amazon Photo (Shchur et al., 2018), and four citation networks: Cora, Pubmed, Citeseer (Sen et al., 2008), and DBLP (Pan et al., 2016). Following Dai et al. (2021), 80% of the nodes are designated for the test set, and 10% for the validation set. For the training set, we randomly select 1% of nodes for the large datasets (Coauthor CS, Amazon Photo, Pubmed, and DBLP), and 5% of nodes for small-scale datasets (Cora and Citeseer) as the labeled nodes. We employ two types of label noise and introduce label noise in the datasets following Yu et al. (2019); Dai et al. (2021): (i) uniform noise, where labels flip to any other class with probability $p/(C-1)$, and (ii) pair noise, where labels only flip to their closest class with probability $p$.

**Baselines.** To show the superiority of our proposed ICGNN, we conduct comparisons with GNNs such as GCN (Kipf & Welling, 2017), as well as other advanced methods designed for noisy labels, namely Forward (Patrini et al., 2017), Coteaching+ (Yu et al., 2019), NRGNN (Dai et al., 2021), RTGNN (Qian et al., 2023), CGNN (Yuan et al., 2023b), CR-GNN (Li et al., 2024b), DND-NET (Ding et al., 2024), and ProCon (Li et al., 2025). For a fair comparison, all methods use GCN as the default backbone.

**Implementation Details.** In the experiments, all baseline methods are re-run under the same settings to ensure a fair comparison. For all datasets and methods, we set the rate of noisy labels to the default value of 20%. For our ICGNN, we assign a teleport probability $\epsilon$ of 0.85 and select $K = 5$ as the number of nearest neighbors. The trade-off hyper-parameter $\alpha$ is set to the default value of 0.5. The maximum number of training epochs is 200. Following Dai et al. (2021), each method is replicated for 5 runs to calculate mean accuracy and standard deviation on the test set for evaluation.

### 4.2. Experimental Results

In this section, we evaluate the performance of our ICGNN along with all baselines for node classification in graphs. The results conducted on six datasets, considering two types of label noises (20% noise rates as default), are presented in Table 1. Based on the quantitative results, we can observe: (i) Classic GNNs (GCN) show poorer performance compared to methods specifically designed for noisy labels, indicating a potential vulnerability to overfitting erroneous labels. (ii) The last six baselines (NRGNN ∼ ProCon) surpass Forward and Coteaching+, highlighting the effectiveness of graph-specific methods in extracting meaningful features and semantics from graph data. Among these methods, DND-NET achieves the best performance by avoiding noise propagation and fully exploiting unlabeled data. (iii) Across all datasets and noise types, our ICGNN consistently attains the highest performance. It attributes to our noise detection via influence contradiction and GMM to effectively identify noisy labels, and ICGNN exploit higher-order structure to learn from unlabeled nodes. (iv) We conduct *Wilcoxon rank-sum tests* to assess statistical significance between our ICGNN and runner-up results on uniform and pair noises, as described in Table 5 of Appendix E. At the 0.1 significance level, the differences between ICGNN and runner-up results are statistically significant in most cases.

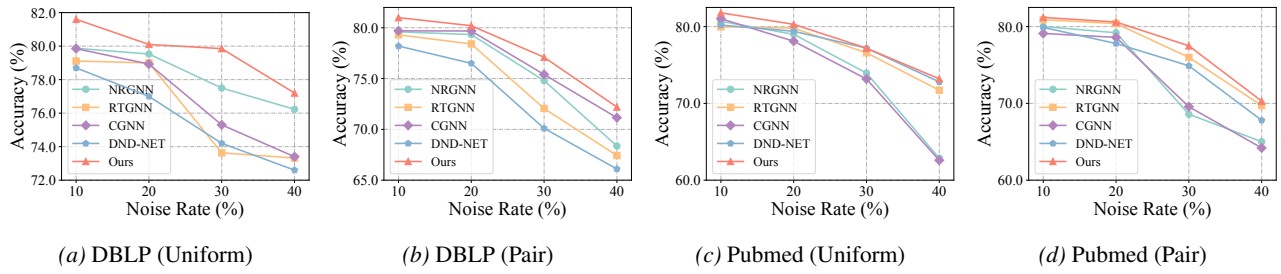

*Figure 2.* Robustness analysis against different levels of label noises on DBLP and Pubmed.

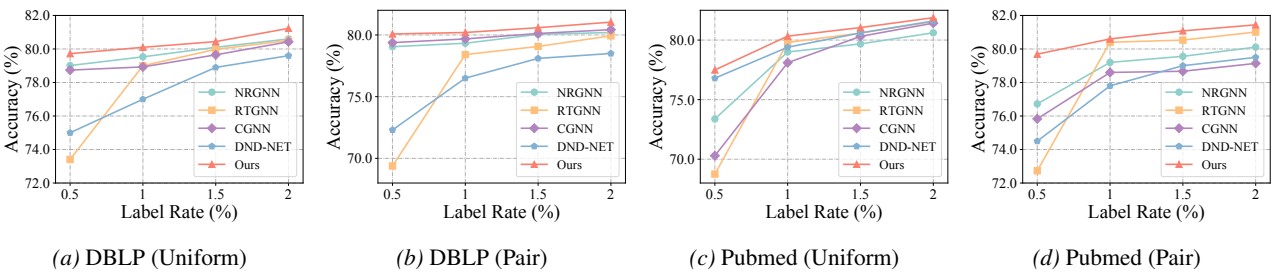

*Figure 3.* The comparison $w.r.t.$ different label rates on DBLP and Pubmed.

*Table 2.* Ablation study against several variants.

| Methods | Cora | | DBLP | |
|---|---|---|---|---|
| | Uniform | Pair | Uniform | Pair |
| ICGNN w/o s-ICS | 79.4±0.9 | 78.1±0.9 | 78.1±1.7 | 79.0±0.6 |
| ICGNN w/o a-ICS | 79.2±1.0 | 77.9±1.1 | 78.3±0.7 | 78.6±0.5 |
| ICGNN w/o NC | 78.7±1.0 | 76.9±1.2 | 77.4±0.8 | 77.1±0.8 |
| ICGNN w/o PL | 79.5±1.2 | 77.2±1.0 | 77.5±1.1 | 78.1±0.4 |
| ICGNN w $\mathbf{A}$ | 79.6±0.9 | 78.2±1.1 | 78.2±1.0 | 78.7±0.6 |
| **ICGNN** | **80.9±0.8** | **79.4±0.7** | **80.1±0.6** | **80.2±0.4** |

## 4.3. Ablation Study

We conduct ablation studies on the Cora and DBLP datasets to show the effectiveness of ICGNN. We examine five variants: (i) ICGNN w/o s-ICS: excludes structure-level ICS; (ii) ICGNN w/o a-ICS: excludes attribute-level ICS; (iii) ICGNN w/o NC: trains a GNN without noise cleaning; (iv) ICGNN w/o PL: removes the pseudo-labeling loss for unlabeled nodes; and (v) ICGNN w $\mathbf{A}$: replaces the graph diffusion matrix $\mathbf{T}$ with the adjacency matrix $\mathbf{A}$.

From Table 2, it is evident that the accuracy drops when either s-ICS or a-ICS is removed. This highlights their complementary nature in capturing the contradiction between nodes in structural and attribute levels, which aids in noise detection and cleaning. We also notice a significant drop in performance when removing the noise cleaning process, which underscores the importance of the noise cleaning process in enhancing the model's robustness against label noise. Moreover, the use of pseudo-labeling loss proves to be beneficial for overall performance by leveraging unlabeled data to provide additional supervision. Finally, we

find that compared to the local information provided by the adjacency matrix $\mathbf{A}$, the global information offered by the graph diffusion matrix $\mathbf{T}$ makes detecting noisy labels through node influence assessment more effective.

## 4.4. Robustness Analysis

To validate the robustness of our ICGNN, we conduct experiments from two perspectives: varying the label noise rate and varying the training label rate. Here we compare our approach against four competitive baselines (NRGNN, RTGNN, CGNN, and DND-NET). Note that CRGNN and ProCon is excluded from the comparison due to its relatively weak performance and unstable fluctuations.

**Impacts of Noisy Label Rates.** To showcase the robust resilience of our ICGNN across various degrees of label noise, we adjust the noise rate in 10%, 20%, 30%, 40%, while maintaining a fixed label rate of 1%. We evaluate the results on DBLP and Pubmed datasets, as reported in Figure 2. With the increase of label noise levels, there is a substantial performance decline across all baseline methods. While our ICGNN also experiences reduced performance, it consistently demonstrates greater resilience in the face of pronounced label noise. This highlights the robust efficacy of our ICGNN in identifying noise through the contradictory influences among nodes and purifying nodes through reliable higher-order neighborhood supervision.

**Impacts of Training Label Rates.** Here we explore the impact of distinct label rates by varying the label rates in 0.5%, 1%, 1.5%, 2%, while keeping both types of noise rates fixed at 20%. The results on DBLP and Pubmed datasets

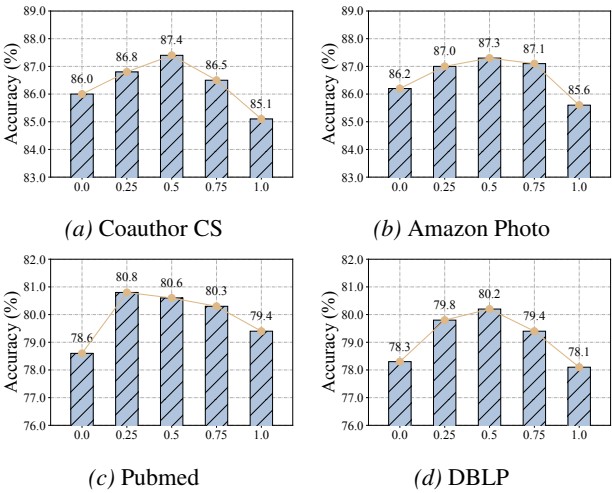

*Figure 4.* Comparison $w.r.t.$ different $\alpha$ on Coauthor CS, Amazon Photo, DBLP, Pubmed datasets.

are shown in Figure 3. With increasing rates, all methods exhibit notable performance improvements owing to the availability of more sufficient supervisory signals. Nevertheless, the efficacy of RTGNN's noise detection through the small-loss criterion might significantly diminish when labeled nodes are scarce, leading to error accumulation during subsequent training and notably low accuracy at a 0.5% label rate. Additionally, our ICGNN consistently surpasses others, even with higher label rates that inevitably involve more noisy labels. This suggests that our proposed ICGNN effectively alleviates the negative effects of a substantial quantity of noisy labels.

### 4.5. Impacts of Hyper-parameter $\alpha$

To verify the relative importance of structural and attributive information in the final ICS value, we analyze the effect of the hyperparameter $\alpha$ in Eq. (5). We set $\alpha$ to values from $\{0, 0.25, 0.5, 0.75, 1.0\}$ and Figure 4 present the results on Coauthor CS, Amazon Photo, DBLP, and Pubmed datasets under uniform noise. It can be seen clearly that relying solely on either structural or attributive information ($\alpha = 0$ or 1) is suboptimal for detecting noisy labels. In most datasets, $\alpha = 0.5$ yields the consistently best results, indicating that both structural and attributive information are crucial and indispensable for noise label detection. Additionally, we observe that their relative importance depends on the dataset's topological density. For example, the Pubmed dataset has a relatively sparse structure compared to the others, leading to better performance with attributive information alone than with structural information, whereas the opposite is true for the other datasets.

Furthermore, we also experiment with defining $\alpha$ as an attention score between the structural and attributive information, making it a learnable parameter. However, as shown in

*Table 3.* Comparison $w.r.t.$ learnable and fixed $\alpha$.

| Uniform Noise | Learned $\alpha$ | $\alpha = 0.5$ |
|---|---|---|
| Coauthor | 87.0±0.7 | **87.4±0.8** |
| Amazon Photo | 86.9±0.5 | **87.3±0.5** |
| Cora | 80.5±0.9 | **80.9±0.8** |
| Pubmed | 79.6±0.8 | **80.3±0.5** |
| DBLP | **80.3±0.7** | 80.1±0.6 |
| Citeseer | 71.1±0.8 | **71.5±0.5** |

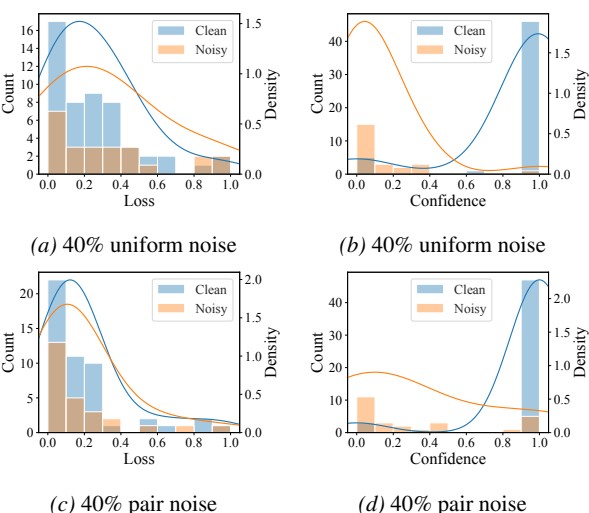

*Figure 5.* Loss distribution (a and c) and confidence distribution (b and d) when training a GCN on Amazon Photo.

Table 3, the results are not as good as using a fixed value, which is why we opt for a fixed value.

### 4.6. Comparisons of Different Noise Indicators

We compare our ICGNN with the widely-used small-loss criterion (Gui et al., 2021) on Amazon Photo dataset to validate the effectiveness of our approach in accurately distinguishing noise in training data. Figure 5a and 5c illustrate histograms and kernel density estimation curves of normalized losses during GCN training at the 50-th epoch. Additionally, we present the distribution of confidences estimated by our ICGNN in Figure 5b and 5d. It is clearly evident that the loss distributions of clean and noisy samples exhibit considerable overlap, underscoring the limited effectiveness of the small-loss criterion in distinguishing between clean and noisy labels. This challenge becomes even more pronounced when pair noise corrupts the dataset. In sharp contrast, the confidences assigned to clean and noisy samples by ICGNN establish well-separated discrete clusters, resulting in a clear and reliable differentiation between them. These results underscore the robust capability of our influence contradiction score and GMM in effectively detecting and mitigating noise.

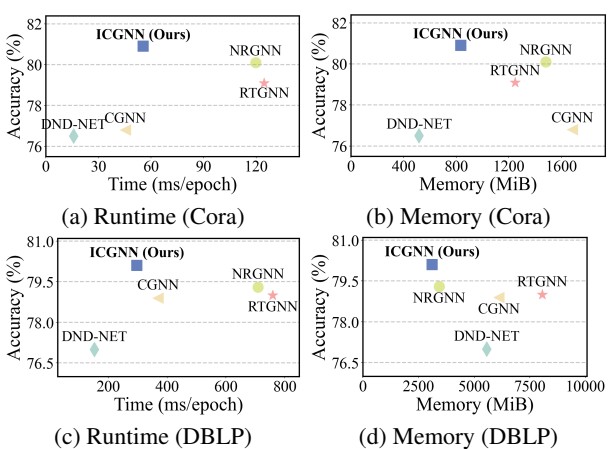

(a) Runtime (Cora)    (b) Memory (Cora)

(c) Runtime (DBLP)    (d) Memory (DBLP)

*Figure 6.* Comparisons of average training time per epoch and memory cost.

*Table 4.* Performance on OGBN-arxiv, Cornell datasets.

| Dataset | OGBN-arxiv | Cornell |
|---|---|---|
| NRGNN | OOM | 40.4±2.0 |
| RTGNN | OOM | 42.3±1.2 |
| CGNN | 21.8 | 31.8±1.2 |
| DND-NET | 25.9 | 38.3±0.9 |
| ICGNN (Ours) | **28.3** | **44.7±1.6** |

### 4.7. Comparison of Runtime and Memory Cost

In this section, we compare the training time and memory cost of different methods to demonstrate efficiency. Let $N$ represents the total number of nodes in the graph, and $L$ represents the number of labeled nodes. For large-scale datasets, $L = 0.01N$, and for small-scale datasets, $L = 0.05N$. Additionally, the complexity of training the GMM using the EM algorithm is very low, i.e., $O(LT)$, where $T$ is the number of EM iterations. In the experiments, when $T$ is set to 10 or fewer, the performance is typically optimal. Therefore, the term $O(LT)$ is omitted in the overall complexity analysis in Appendix C, as it is negligible.

Here we provide a detailed comparison of the training time in millisecond (ms) and memory consumption in mega bytes (MiB) for our ICGNN and the competitive baselines (NRGNN, RTGNN, CGNN and DND-NET). As shown in Figure 6, our method achieves higher computational efficiency and comparable memory consumption while maintaining excellent performance.

### 4.8. Performance on Large-scale and Heterophilous Datasets

In addition to the six benchmarks considered in the main experiments, we add a large-scale dataset OGBN-Arxiv, a heterophilous network dataset Cornell, to demonstrate

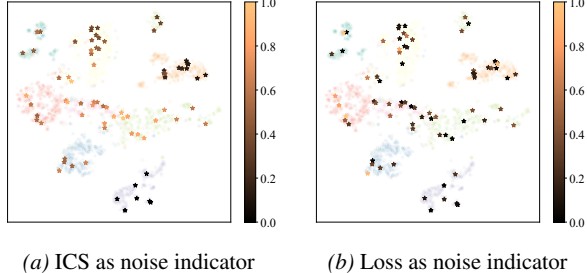

*(a)* ICS as noise indicator    *(b)* Loss as noise indicator

*Figure 7.* Effectiveness of influence contradiction score as noise indicator (t-SNE visualization of the Amazon Photo dataset with 40% uniform noise ).

the broad scalability and generalizability of our proposed ICGNN. The experimental results under uniform noise are shown in Table 4. On OGBN-Arxiv dataset, it can be observed that both NRGNN and RTGNN suffer from direct memory allocation issues, while our ICGNN significantly outperforms CGNN and DND-NET. It demonstrates the flexibility, effectiveness, and scalability of our approach. On the heterophilous dataset Cornell, compared to highly competitive baselines, our method ICGNN still achieves the best performance, which further verify the broad applicability across different types of graphs.

### 4.9. Visualization Analysis

We present a visual representation of node embeddings and their corresponding labels using t-SNE in Figure 7. In addition, we color the nodes in the training set differently based on two distinct noise indicators: ICS and loss. As shown in Figure 7a, nodes with lower influence contradiction (darker color) are positioned farther away from class boundaries compared to nodes with higher conflict (lighter color), highlighting the fact that nodes close to class boundaries are more prone to be noisy. On the contrary, when employing loss as the noise indicator (Figure 7b), the differentiation between clean and noisy nodes is less significant. The findings emphasize that ICS excels in revealing conflicts between nodes at both the structural and attribute levels, making it a more suitable choice as the noise indicator.

## 5. Conclusion

In this study, we present a robust GNN named ICGNN to handle noisy and limited labels. To effectively detect noisy labels on the graph, we design a noise indicator to measure the influence contradiction score from both structure- and attribute-level. Moreover, we develop a soft strategy to cautiously correct detected noisy labels by combining predictions from neighboring nodes. Pseudo labels are also generated for unlabeled nodes to further provide sufficient supervision signals. Empirical studies on multiple datasets confirm the effectiveness of our ICGNN against label noise.

## Acknowledgment

This work is supported in part by the National Key Research and Development Program of China with Grant No. 2023YFC3341203, the National Natural Science Foundation of China under Grant 62276002, 12501344 and 62306014, Postdoctoral Fellowship Program (Grade A) of CPSF under Grant BX20240239 and BX20250376, China Postdoctoral Science Foundation under Grant 2024M762201, Sichuan Science and Technology Program under Grant 2025ZNSFSC0808 and 2025ZNSFSC1506, the Fundamental Research Funds for the Central Universities under Grant 1082204112K97, and Sichuan University Interdisciplinary Innovation Fund.

## Impact Statement

The proposed ICGNN presents a valuable contribution to machine learning, particularly in addressing label noise in GNNs. ICGNN introduces a noise indicator based on the influence contradiction score and a Gaussian mixture model to detect and correct noisy labels. This improves GNN accuracy in real-world scenarios and reduces biases from incorrect annotations, making predictions more reliable. By enhancing robustness against label noise, our ICGNN helps develop more dependable machine learning models.

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

# A. Proof of Theorem 3.1

Let $\mathbf{M}_{ki} = \alpha\mathbf{T}_{ki} + (1-\alpha)\mathbf{R}_{ki}$. Due to column normalization, we partition the sum by class and yield:

$$S_i = \sum_{j=1}^{C} \frac{1}{|\mathcal{C}_j|} \sum_{k\in\mathcal{C}_j} \mathbf{M}_{ki} = \sum_{k=1}^{N} \frac{\mathbf{M}_{ki}}{|\mathcal{C}_{y_k}|} \leq \sum_{k=1}^{N} \mathbf{M}_{ki} = 1,$$

where the inequality follows from $\frac{1}{|C_{y_k}|} \leq 1$ (each class contains at least one node). By the definition of ICS, we have:

$$\text{ICS}_i = \sum_{j\neq y_i} \frac{1}{|\mathcal{C}_j|} \sum_{k\in\mathcal{C}_j} \mathbf{M}_{ki} = S_i - \frac{1}{|\mathcal{C}_{y_i}|} \sum_{k\in\mathcal{C}_{y_i}} \mathbf{M}_{ki}.$$

For the case where $y_i = y_i^*$ (clean node), we have:

$$\text{ICS}_i = S_i - \frac{1}{|C_{y_i^*}|} \sum_{k\in C_{y_i^*}} \mathbf{M}_{ki}.$$

By the homophily assumption, $\frac{1}{|C_{y_i^*}|} \sum_{k\in C_{y_i^*}} M_{ki} \geq \delta$, and together with $S_i \leq 1$, we obtain $\text{ICS}_i \leq 1 - \delta$.

For the case where $y_i \neq y_i^*$ (noisy node), the term $j = y_i^*$ is included in the sum $\sum_{j\neq y_i}$. Therefore, we have:

$$\text{ICS}_i \geq \frac{1}{|C_{y_i^*}|} \sum_{k\in C_{y_i^*}} \mathbf{M}_{ki} \geq \delta.$$

# B. Further Discussion on our proposed ICGNN

### B.1. Handling of noisy nodes difficult to detect by ICS

In situations where it is difficult to determine whether there is noise through ICS, it is more likely to occur near class boundaries, where samples from different categories are closely distributed and harder to separate. In these regions, the challenge lies in accurately distinguishing whether the observed noise is due to mislabeling or if it naturally arises from the intrinsic ambiguity of the class boundary. A potentially effective solution is to incorporate local structural information (such as node degree, random walks, etc.) or to synthesize virtual nodes to make the decision boundaries between different classes more distinct. It could enhance the effectiveness of our detection strategy in distinguishing whether a label is noisy.

### B.2. Gradient propagation of ICGNN

In the implementation of ICGNN, noise detection and label correction are implemented outside the direct gradient computation path to ensure that the primary model's training process remains entirely unaffected. Specifically, ICS calculation, GMM-based noise confidence assignment, and the overall label correction process are all designed as auxiliary operations to iteratively refine the training labels and provide improved supervision by accurately identifying and adjusting noisy labels. These operations do not backpropagate gradients to the model parameters. Instead, they act as a separate preprocessing step that dynamically adjusts the labels used for training while keeping the gradient flow strictly intact through the primary model architecture. This design ensures that the detection and correction mechanisms do not interfere with the model's gradient-based optimization process.

### B.3. Label Noise *v.s.* Hard/Unusual Nodes

Our proposed ICS detects label noise from the perspective of influence contradiction at both the structural and attribute levels. Hard or unusual nodes typically show contradiction in only one dimension: hard nodes have ambiguous attributes but consistent neighbors; heterophilous nodes have cross-class connections but consistent attributes. Therefore, ICS is generally able to distinguish mislabeled nodes from hard or unusual nodes, because a node is detected as noisy by ICS only when both structural and attribute contradictions are strong. Furthermore, Section 4.9 provides an experimental analysis of ICS's effectiveness in identifying noisy samples. Figure 7a shows lower ICS nodes lie farther from class boundaries, indicating our method effectively isolates clean nodes (as boundary nodes are more prone to noise). Figure 7b shows loss fails to provide clear separation, as high-loss nodes do not consistently align with class boundaries. ICS provides a nuanced view of node consistency that goes beyond what the loss metric can capture.

## B.4. Effectiveness on Heterophilous Graphs

Our method does not simply rely on the structure homophily assumption; instead, it identifies label noise by leveraging contradictions in both structural and attribute information. In heterophilous graphs, attribute features often align with true class even when structure does not. Thus, our ICS can alleviate interference from heterophilous graph structures: a large ICS value arises only when both structural and attribute contradictions are substantial. Even when ICS is affected by heterophily, our GMM-guided soft-threshold mechanism avoids discarding correct information, mitigating misclassification impact. Empirically, we show the performance comparison between our ICGNN and competitive baselines on the heterophilous graph Cornell in Table 4, where ICGNN demonstrates good applicability across different types of graphs.

# C. Complexity Analysis

Assume the edge number is $E$ and the iteration number in fitting GMM is $T$. We compute the ICSs in $O(EN + NL + L^3)$ time. Since $L \ll N$, the process does not consume too much time. Based on ICSs, the complexities of training GMM, label cleaning, and calculating loss are $O(LT)$, $O(CN)$, and $O(CN)$, respectively. Hence, the overall complexity of ICGNN is $O(EN + L^3 + CN)$, which scales linearly with the sample size.

In fully-supervised settings, the $O(L^3)$ term mainly originates from the computation of the diffusion matrix in Eq. (4). To reduce the complexity, we can first use low-precision methods (such as approximating the diffusion matrix via power iteration or Monte Carlo methods, and reducing the number of iterations in power iteration or the number of walks in Monte Carlo methods) to quickly screen candidate noisy nodes, and then perform high-precision computation on the candidate set. Additionally, high-confidence nodes can reuse historical ICS values, updating only low-confidence nodes, which further reduces the computational cost in subsequent epochs. Moreover, we can directly sparsify the adjacency matrix through sampling or truncation, thereby reducing the complexity of diffusion matrix computation to linear (Klicpera et al., 2019).

# D. Pseudo-Code of Our Framework

The whole optimization process of our proposed ICGNN is summarized in Algorithm 1.

---

**Algorithm 1** The Optimization Algorithm of ICGNN

---

**Input**: Graph $\mathcal{G} = \{\mathcal{V}, \mathbf{A}, \mathbf{X}, \mathbf{Y}_{\mathrm{L}}\}$; Maximum number of iterations $I_{\max}$.
**Output**: Predictions on the unlabeled nodes in $\mathcal{V}_{\mathrm{U}}$.

1: Initialize the trainable parameters in the GNN encoder;
2: Calculate the structure-level ICS values: $\mathrm{ICS}_i(\mathbf{T}), i = 1, \ldots, L$;
3: Set $t = 0$;
4: **while** $t \leq I_{\max}$ **do**
5:    Update the node representations $\{\mathbf{Z}_{\mathrm{L}}, \mathbf{Z}_{\mathrm{U}}\}$ from the GNN-based encoder;
6:    Calculate the ICS values by Eq. (5);
7:    Implement the EM algorithm to achieve the confidence $\hat{\beta}_i^{(t)}, i = 1, \ldots, L$;
8:    Update the labels for the labeled nodes by Eq. (6);
9:    Annotate pseudo-labels for unlabeled nodes by Eq. (7);
10:    Calculate the total loss $\mathcal{L}$ by Eq. (8);
11:    Conduct back-propagation and update the whole network in ICGNN by minimizing $\mathcal{L}$;
12:    $t = t + 1$;
13: **end while**
14: Obtain the labels of unlabeled nodes in $\mathcal{V}_{\mathrm{U}}$ by making predictions based on the learned representations $\mathbf{Z}_{\mathrm{U}}$.

---

# E. Extra Experimental Analysis

## E.1. Significance Testing of Experimental Results

To assess the statistical significance of the performance differences, we perform Wilcoxon rank-sum tests comparing our ICGNN with the runner-up methods. The analysis, detailed in Table 5, is conducted on results from uniform and pair noise settings. Specifically, at the 0.1 significance level, the null hypothesis is rejected in the vast majority of test cases, indicating that the advantages of ICGNN over the runner-up are statistically significant and not merely due to random chance.

*Table 5.* Statistical significance.

| Datasets | p-value | Datasets | p-value |
|---|---|---|---|
| Coauthor CS | 0.0473 | Pubmed | 0.1474 |
| Amazon Photo | 0.0061 | DBLP | 0.0718 |
| Cora | 0.0468 | Citeseer | 0.0712 |

(a) Uniform Noise

| Datasets | p-value | Datasets | p-value |
|---|---|---|---|
| Coauthor CS | 0.0718 | Pubmed | 0.1050 |
| Amazon Photo | 0.0184 | DBLP | 0.0718 |
| Cora | 0.0468 | Citeseer | 0.0108 |

(b) Pair Noise

## E.2. Performance evaluation on different GNNs

*Table 6.* Performance of ICGNN with different GNN backbones.

| | Coauthor CS | Amazon Photo | Cora | Pubmed | DBLP | Citeseer |
|---|---|---|---|---|---|---|
| GCN | **87.4±0.8** | 87.3±0.5 | **80.9±0.8** | **80.3±0.5** | 80.1±0.6 | **71.5±0.5** |
| GAT | 86.8±1.2 | **87.4±0.7** | 80.2±1.1 | 80.1±0.7 | 79.8±0.7 | 71.0±0.7 |
| GIN | 87.1±0.7 | 87.0±0.6 | 80.6±0.9 | 79.9±0.5 | **80.3±0.6** | 71.3±0.4 |

Note that our method ICGNN does not use the graph diffusion within a specific GNN but rather employs it as a tool to capture finer global neighbor relationships for designing noisy label detection criterion and correction strategies. As such, our proposed method is GNN-agnostic, allowing users to substitute any GNN model to adapt our approach.

In the main experiments, our method is implemented with GCN (Kipf & Welling, 2017) as the backbone. Additionally, we include two other GNN variants (GAT (Veličković et al., 2018) and GIN (Xu et al., 2018)) under uniform noise for comparison, shown in Table 6. The performance fluctuations across different GNN variants in our method are relatively small, which demonstrates the robustness of our method to various GNN architectures. Moreover, GCN achieves the best results on most datasets, which explains our choice of GCN as the backbone.

## E.3. Robustness analysis against different levels of label noises

To illustrate the robustness of our proposed ICGNN under varying degrees of label noise, we systematically adjust the noise rate in increments of {10%, 20%, 30%, 40%}, while maintaining a fixed label rate of 1%. Our evaluation focuses on comparing the performance of our ICGNN against leading baselines (NRGNN, RTGNN, CGNN and DND-NET) across four datasets (Coauthor CS, Amazon Photo, Cora and Citeseer). The experimental results are depicted in Figure 8.

From the figure, we can see a noticeable decline in performance across all baseline methods as the level of noise rates increase. While ICGNN also experiences a reduction in performance, it distinguishes itself by exhibiting remarkable resilience in the presence of substantial label noise. Notably, as the label noise intensifies, the performance gap between ICGNN and the baselines widens, underscoring the effectiveness of our proposed approach. This observed resilience is primarily attributed to ICGNN's remarkable ability to identify and mitigate noise through the intricate interplay of contradictory influences among nodes. The incorporation of higher-order neighborhood supervision further enhances the purification process, solidifying the efficacy of our proposed ICGNN in navigating and mitigating the impact of label noise.

## E.4. Sensitivity analysis against different label rates

In this part, we investigate the impact of varying label rates by manipulating the label rates within {0.5%, 1%, 1.5%, 2%} for Coauthor CS and Amazon Photo datasets, and {2.5%, 5%, 7.5%, 10%} for Cora and Citeseer datasets, while maintaining a fixed 20% for both types of noise rates. The results, shown in Figure 9, provide insights into the performance variations of different methods under these conditions, evaluated on the Coauthor CS, Amazon Photo, Cora, and Citeseer datasets.

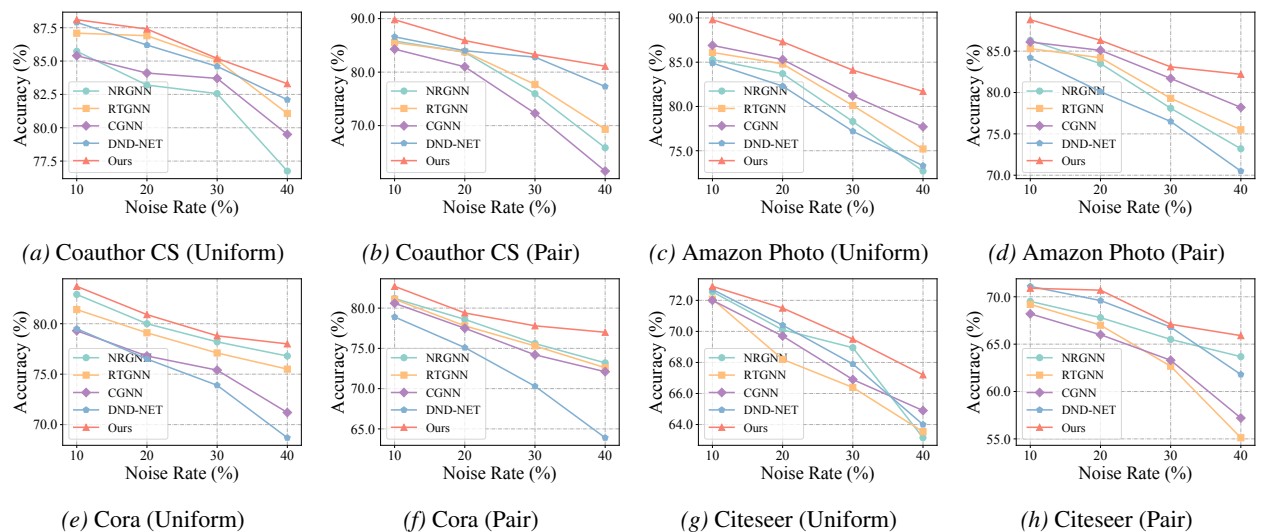

*Figure 8.* Robustness analysis against different levels of label noises on four datasets.

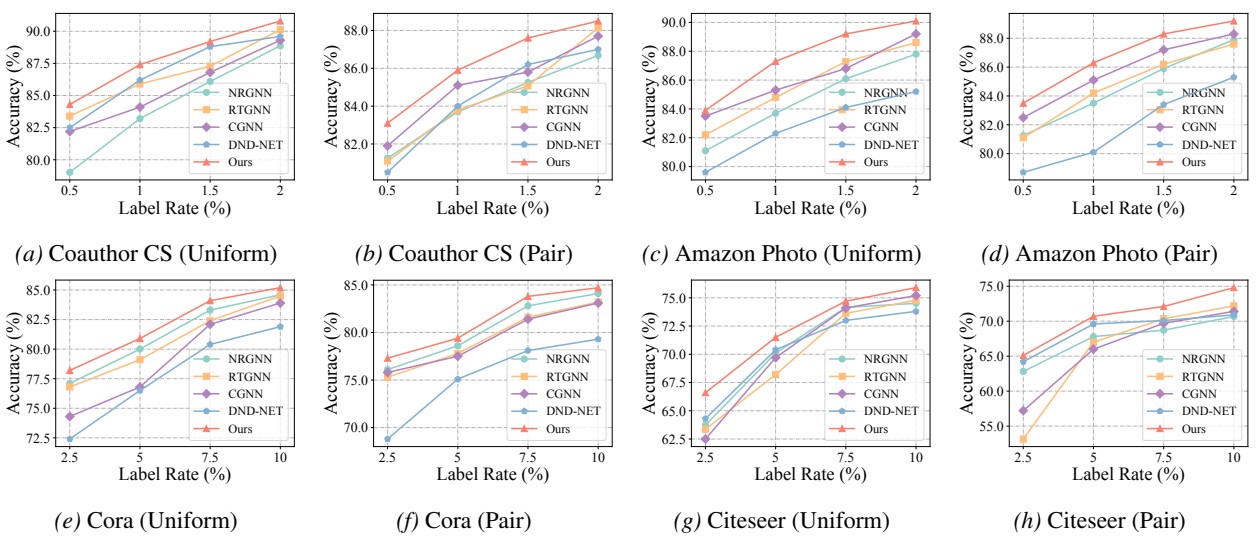

*Figure 9.* Sensitivity analysis against different label rates on four datasets.

As label rates increase, the performance of all methods improves significantly due to the increased availability of supervison signals. Remarkably, NRGNN consistently outperforms RTGNN and CGNN, especially in scenarios characterized by a scarcity of labeled nodes (0.5%). It underscores the effectiveness of NRGNN in generating accurate pseudo-labels, thereby augmenting the overall supervision. Interestingly, the efficacy of RTGNN in noise detection, reliant on the small-loss criterion, appears to diminish when labeled nodes are sparse, resulting in error accumulation during subsequent training and notably diminished accuracy at a 0.5% label rate.

In contrast, our ICGNN consistently surpasses other methods, even when confronted with higher label rates that entail a greater proportion of noisy labels. This compelling performance suggests that our ICGNN effectively mitigates the adverse effects associated with a substantial quantity of noisy labels via our effective detection mechanism and correction strategy.

### E.5. Extreme scenarios with high noise rates

To verify the robust performance of our method ICGNN in more extreme scenarios, we include the results of our method and three competitive baseline methods on the DBLP dataset under uniform noise levels of 60% and 80% in Table 7.

It is evident that when the noise ratio exceeds 50%, the performance of all methods drops significantly. However, our method still maintains a certain degree of superior robustness compared to the baselines. At the same time, we realize that when the noise ratio becomes excessively high, utilizing these incorrect label information may do more harm than good. In such cases, it is crucial to extract more effective discriminative information from the data itself for learning node representations.

*Table 7.* Performance on DBLP under uniform noise levels of 60% and 80%.

| Noise Rate | 60% | 80% |
|---|---|---|
| NRGNN | 42.2±3.1 | 24.9±4.0 |
| RTGNN | 45.3±2.9 | 20.2±4.5 |
| CGNN | 41.7±4.6 | 19.8±2.4 |
| DND-NET | 44.6±3.4 | 24.4±4.5 |
| ICGNN w/o PL | 43.6±2.5 | 23.6±3.1 |
| ICGNN (Ours) | **47.0±2.7** | **26.0±3.2** |

In addition, we also test the performance of the pseudo-labeling module in high-noise scenarios. We include a variant without pseudo-label loss on the DBLP dataset in the second-to-last row of Table 7. When the noise ratio becomes too high, although the pseudo-labeling technique inevitably generates incorrect labels, we observed that removing the pseudo-label loss results in even worse performance. It demonstrates the effectiveness of our pseudo-labeling approach. Furthermore, our proposed pseudo-labeling technique relies on predictions from neighboring nodes, rather than directly from the target node, which effectively helps mitigate the negative impact of incorrect labels being introduced.

## F. Performance Comparisons with more approaches

To more comprehensively validate the superiority of our proposed ICGNN, we conduct comparisons on three datasets (Cora, Pubmed, Citeseer) against 12 additional competitive approaches, including six Learning with Label Noise (LLN) methods (Backward (Patrini et al., 2017), Coteaching (Han et al., 2018), SCE (Wang et al., 2019), JoCoR (Wei et al., 2020), APL (Ma et al., 2020), and S-model (Goldberger & Ben-Reuven, 2017)) and six Graph Neural Networks under Label Noise (GLN) methods (CP (Zhang et al., 2020), D-GNN (NT et al., 2019), UnionNET (Li et al., 2021), CLNode (Wei et al., 2023), PIGNN (Du et al., 2023), and RNCGLN (Zhu et al., 2024)). The experimental results are shown in Table 8 and Table 9.

*Table 8.* Performance on three datasets (mean±std). The best results in all the methods are highlighted with **bold**. The noise rate is set to 20% as default.

| | Methods | Backward | Coteaching | SCE | JoCoR | APL | S-model | ICGNN |
|---|---|---|---|---|---|---|---|---|
| **Uniform** | Cora | 75.9±1.5 | 66.7±4.2 | 76.1±1.4 | 76.3±1.7 | 76.0±1.6 | 75.9±1.3 | **80.9±0.8** |
| | Pubmed | 69.7±3.9 | 68.9±2.9 | 71.3±3.2 | 61.2±6.9 | 70.9±3.4 | 70.5±3.6 | **80.3±0.5** |
| | Citeseer | 62.4±2.6 | 50.9±4.2 | 62.5±2.9 | 65.9±2.5 | 60.7±2.6 | 61.1±3.1 | **71.5±0.5** |
| **Pair** | Cora | 73.1±3.2 | 64.6±2.6 | 73.7±1.9 | 71.5±6.8 | 73.6±2.2 | 73.0±2.3 | **79.4±0.7** |
| | Pubmed | 71.0±6.4 | 68.6±3.8 | 72.1±5.2 | 61.8±7.4 | 71.1±6.0 | 70.8±6.7 | **80.6±0.3** |
| | Citeseer | 58.5±3.4 | 50.7±4.7 | 58.9±3.2 | 61.1±5.6 | 56.7±4.4 | 57.8±3.7 | **70.7±0.7** |

From the above tables, it can be clearly observed that our method consistently outperforms both categories (LLN methods and GLN methods) across all datasets, which demonstrates its effectiveness in handling noisy labels in graph data. By detecting noisy labels through the influence contradiction score and GMM, and further applying a neighbor-based soft correction strategy, our approach maximally alleviates the adverse effects of incorrect labels during training.

## G. Related Work

### G.1. Graph Neural Networks

GNNs have gained remarkable success and popularity in various domains, and the research landscape can be broadly categorized into two primary directions: spectral-based and spatial-based (Ju et al., 2024a). For spectral-based methods,

*Table 9.* Performance on three datasets (mean±std). The best results in all the methods are highlighted with **bold**. The noise rate is set to 20% as default.

| | Methods | CP | D-GNN | UnionNET | CLNode | PIGNN | RNCGLN | ICGNN |
|---|---|---|---|---|---|---|---|---|
| **Uniform** | Cora | 76.7±1.5 | 64.7±4.0 | 76.1±1.7 | 73.5±1.9 | 74.1±2.0 | 76.9±1.2 | **80.9±0.8** |
| | Pubmed | 68.4±9.0 | 65.2±4.4 | 70.2±3.9 | 67.7±3.8 | 71.8±2.4 | N/A | **80.3±0.5** |
| | Citeseer | 61.4±3.0 | 52.2±3.6 | 66.5±3.6 | 59.6±3.2 | 64.1±1.7 | 65.3±4.4 | **71.5±0.5** |
| **Pair** | Cora | 72.7±3.3 | 64.6±3.1 | 73.0±3.0 | 71.8±1.5 | 70.7±1.3 | 75.3±2.8 | **79.4±0.7** |
| | Pubmed | 68.6±4.4 | 67.2±4.0 | 71.4±6.6 | 69.6±7.1 | 72.1±5.0 | N/A | **80.6±0.3** |
| | Citeseer | 57.3±3.0 | 51.5±3.4 | 61.5±5.0 | 58.4±4.3 | 61.3±4.0 | 61.1±7.2 | **70.7±0.7** |

researchers have focused on leveraging graph Laplacians or graph Fourier transforms to embed nodes in a lower-dimensional space. For example, ChebNet (Defferrard et al., 2016) utilizes Chebyshev polynomials to efficiently approximate graph convolutions and capture spectral information, enabling the model to learn meaningful representations of nodes in the graph. In contrast, spatial-based methods involve GNNs that directly process node feature representations and their neighbors, enabling localized message passing (Gilmer et al., 2017). Benefiting from their excellent performance, GNNs have found extensive applications in tasks such as node classification (Luo et al., 2023a; Yi et al., 2025; Wen et al., 2025), link prediction (Subramonian et al., 2023; Shi et al., 2024), graph clustering (Liang et al., 2025; Zhang et al., 2026; Ju et al., 2026), and graph classification (Luo et al., 2023b; Mao et al., 2023; Wang et al., 2024a). However, GNNs typically assume a clean annotation environment, and still struggle with handling noisy and limited labels, while our ICGNN overcomes these issues by designing an effective noise indicator and developing a robust correction strategy against label noise and scarcity.

## G.2. Neural Networks with Noisy Labels

Deep learning powered by neural networks has achieved impressive performance across diverse domains. However, the notorious issue of noisy labels poses a significant challenge to their efficacy (Zhang et al., 2021). To tackle this, various approaches in vision domains have been proposed to address the challenge of noisy labels, which can be broadly categorized into three classes: *sample selection* (Han et al., 2018; Yu et al., 2019; Li et al., 2020; Pan et al., 2025), *loss correction* (Ghosh et al., 2017; Wang et al., 2019; Zhang & Sabuncu, 2018; Wilton & Ye, 2024; Nagaraj et al., 2025), and *label correction* (Sheng et al., 2017; Song et al., 2019; Li et al., 2024a). For example, as a representative work, Co-teaching (Han et al., 2018) employs two networks that are trained to identify clean samples, and iteratively exchange and refine each other. Nagaraj et al. (2025) introduce the problem of temporal label noise in time series classification and develop methods that estimate time-dependent noise functions to train more robust classifiers. However, these methods encounter obstacles when applied to graph data due to the intricate graph structures. To address these issues, recently there are a handful of algorithms have been proposed to address noisy labels on graphs (Dai et al., 2021; 2022; Li et al., 2024b; Qian et al., 2023; Xia et al., 2023; Yuan et al., 2023b; Ding et al., 2024; Li et al., 2025; Zhao et al., 2026). Grounded in these advanced works, Wang et al. (2024b) and Kim et al. (2025) introduce comprehensive benchmarks for GNNs under label noise, enabling fair comparisons and yielding new insights for future research. Among these competitive approaches, PI-GNN (Du et al., 2023) proposes a pairwise framework for noisy node classification, combining confidence-aware PI estimation with decoupled training to enhance robustness via pairwise node interactions. ERASE (Chen et al., 2023) enhances label noise tolerance via structural denoising and decoupled label propagation, combining prototype pseudo-labels with denoised labels for robust node classification. DND-NET (Ding et al., 2024) introduces a noise-robust GNN that avoids label noise propagation and a reliable pseudo-labeling algorithm to leverage unlabeled nodes while mitigating noise effects. Wu et al. (2024) and Cheng et al. (2024) further extend learning against label noise to heterophilic graphs, demonstrating their effectiveness under challenging graph structures. However, these approaches struggle to effectively detect whether a node is noisy and lack a robust algorithm for label correction. Our framework ICGNN goes further and develops an effective detection approach as well as a rational correction strategy to enhance the robustness of GNNs.

# H. Dataset Details

For our comprehensive evaluation, we employ six datasets across diverse domains. These datasets encompass a range of network types, such as an author network Coauthor CS (Shchur et al., 2018), a co-purchase network Amazon Photo (Shchur et al., 2018), and four citation networks: Cora, Pubmed, Citeseer (Sen et al., 2008), and DBLP (Pan et al., 2016).

**Coauthor CS** (Shchur et al., 2018): This dataset represents a co-authorship network in the field of computer science. Nodes typically represent authors, and edges indicate collaboration between authors on scholarly publications.

**Amazon Photo** (Shchur et al., 2018): This dataset consists of product images and metadata from Amazon, focused on photo-related products. It captures user interactions, including product co-purchases, and is used for tasks like recommendation and item classification in e-commerce.

**Cora** (Sen et al., 2008): It is a citation network derived from a computer science paper repository. Nodes represent papers, and edges denote citations between them. It serves as a common benchmark for evaluating graph-based algorithms in information retrieval and recommender systems.

**Pubmed** (Sen et al., 2008): Similar to Cora, Pubmed is another citation network originating from the biomedical domain. Nodes represent scientific articles, and edges signify citations. It is widely used in research for evaluating algorithms in the biomedical and healthcare domains.

**Citeseer** (Sen et al., 2008): It consists of scientific publications in computer science, with each paper represented as a node. The dataset includes citation relationships between papers, and the features represent word occurrences, widely used for graph-based classification tasks.

**DBLP** (Pan et al., 2016): This dataset is a citation network encompassing computer science and related fields. Nodes represent publications, and edges represent citations. It is a widely used dataset for evaluating algorithms in bibliographic analysis and citation recommendation.

## I. Baseline Details

To showcase the effectiveness of our proposed ICGNN, we perform comprehensive comparisons with several state-of-the-art GNN-based methods. This includes well-established GNNs such as GCN (Kipf & Welling, 2017). Additionally, we evaluate our method against other models specifically designed to handle noisy labels, namely Forward (Patrini et al., 2017), Coteaching+ (Yu et al., 2019), NRGNN (Dai et al., 2021), RTGNN (Qian et al., 2023), CGNN (Yuan et al., 2023b), CR-GNN (Li et al., 2024b), DND-NET (Ding et al., 2024), and ProCon (Li et al., 2025).

**GCN** (Kipf & Welling, 2017): It is a foundational graph neural network architecture widely adopted. It leverages graph convolutional layers to capture node representations by aggregating information from neighboring nodes.

**Forward** (Patrini et al., 2017): This method is designed to address noisy labels by employing a forward correction mechanism during training. It iteratively updates the estimated labels to minimize the impact of noisy annotations.

**Coteaching+** (Yu et al., 2019): This method is a noise-robust training strategy that involves two networks, each learning from the other's more confident predictions. It aims to reduce the influence of noisy labels during training.

**NRGNN** (Dai et al., 2021): This method learns a robust GNN with noisy, limited labels by linking unlabeled nodes to labeled ones with high feature similarity, providing clean labels and generating pseudo labels for extra supervision.

**RTGNN** (Qian et al., 2023): This method governs label noise by adaptively applying self-reinforcement and consistency regularization, correcting noisy labels and generating pseudo-labels to focus on clean labels while reducing noisy ones.

**CGNN** (Yuan et al., 2023b): This method employs graph contrastive learning and a homophily-based sample selection technique to enhance the robustness of node representations against label noise and purify noisy labels for efficient graph learning.

**CR-GNN** (Li et al., 2024b): This method tackles sparse and noisy labels by integrating neighbor contrastive loss, a dynamic cross-entropy loss that selects reliable nodes, and a cross-space consistency constraint to enhance robustness.

**DND-NET** (Ding et al., 2024): This method develops a simple yet effective label noise propagation-free GNN backbone and a novel reliable graph pseudo-labeling algorithm to prevent overfitting and leverage unlabeled nodes.

**ProCon** (Li et al., 2025): This method identifies mislabeled nodes by measuring their label consistency with semantically similar peers and employs a Gaussian Mixture Model to distinguish clean samples, which iteratively refines the prototypes for improved detection.

