# OpenReview forum: "Identifying and Correcting Label Noise for Robust GNNs via Influence Contradiction"
_ICML.cc/2026/Conference — ICML 2026 regular_

### Official Review · Reviewer_ynSy · 2026-03-03

**Soundness:** 4
**Presentation:** 3
**Significance:** 3
**Originality:** 4
**Overall Recommendation:** 5
**Confidence:** 5

**Summary:**

This paper proposes a robust graph neural network for handling label noise in semi-supervised node classification. The model estimates conflicting scores by integrating structural and attribute features, and employs a Gaussian mixture model to detect noisy labels. On this basis, a soft neighbor aggregation strategy is applied for label correction, while pseudo-labels are generated for unlabeled nodes to enhance the supervision signal. Extensive experiments on six benchmark datasets as well as large-scale and heterogeneous graphs demonstrate the superior performance of the proposed method.

**Compliance With Llm Reviewing Policy:**

Affirmed.

**Final Justification:**

The author's detailed explanation further addressed my concerns, and I am very satisfied with it, so I have raised my score accordingly.

**Key Questions For Authors:**

- If the neighbor nodes of a target node have noisy labels, what impact might this have on detection?
- Some recent studies on noisy labels could be considered for comparison, such as RNCGLN, PIGNN, and CLNode.
- The authors use GCN for graph structure encoding. If other GNN architectures were adopted, how might this affect the results?
- Some spelling errors in individual words need to be checked.

**Limitations:**

Yes

**Strengths And Weaknesses:**

## Strengths
- The paper is well-written and easy to follow.
- The label detection mechanism proposed by the authors is highly innovative, incorporating structural guidance to enhance the rationality and accuracy of detection.
- Comprehensive experiments demonstrate the effectiveness of the proposed approach, particularly on large-scale and heterogeneous graphs, highlighting its generalizability.
- Additionally, the analysis of runtime and memory consumption further validates the efficiency of the model.

## Weaknesses
- Noisy neighbor nodes may weaken the detection of ICS.
- Some baseline methods may need to be supplemented for comparison.
- The impact of different GNN architectures on the results requires further analysis.

---

> ### Author Rebuttal · Authors · 2026-03-31
>
> We appreciate your thoughtful comments and suggestions. We will make every effort to address your concerns as follows.
>
> > **Q1**: Noisy neighbor nodes may weaken the detection of ICS. If the neighbor nodes of a target node have noisy labels, what impact might this have on detection?
>
> **R**: Thanks for your valuable question! Our method is not prone to introducing noisy neighbor information. In Eq. (7), we use the predicted class assignments of neighboring nodes rather than their true labels. It helps to avoid introducing noisy labels during the label correction process to some extent. By utilizing the predicted assignment information, we also enhance the utilization of information from unlabeled nodes.
>
> This strategy is further supported by the principle that nodes with stronger connectivity or higher similarity are more likely to share the same label. Therefore, using predicted assignments from neighboring nodes provides a reliable and robust basis for correction. In this sense, the process is essentially self-supervised and self-enhancing: the model iteratively refines its own predictions to guide supervision. As a result, effective optimization is achieved while mitigating the negative impact of noisy labels.
>
> > **Q2**: Some recent studies on noisy labels could be considered for comparison, such as RNCGLN, PIGNN, and CLNode.
>
> **R**: We appreciate the valuable suggestion. We have compared our method ICGNN with RNCGLN, PIGNN, and CLNode. The results are reported in the table below, which demonstrate that our approach still exhibits superior performance relative to these baselines.
>
> | Uniform Noise  | Cora | Pubmed |  Citeseer |
> | --- | --- | --- | --- |
> | RNCGLN  | 76.9 ± 1.2 | N/A | 65.3 ± 4.4 |
> | PIGNN  | 74.1 ± 2.0 | 71.8 ± 2.4 | 64.1 ± 1.7 |
> | CLNode  | 73.5 ± 1.9 | 67.7 ± 3.8 | 59.6 ± 3.2 |
> | ICGNN  | **80.9 ± 0.8** | **80.3 ± 0.5** |  **71.5 ± 0.5** |
>
> We will incorporate these results into the revised version.
>
> > **Q3**: The authors use GCN for graph structure encoding. If other GNN architectures were adopted, how might this affect the results?
>
> **R**: Thank you for your question. In Appendix D.1, we present two other GNN variants (GAT and GIN) under uniform noise for comparison (as shown in the table below). The performance fluctuations across different GNN variants in our method are relatively small, which demonstrates the robustness of our method to various GNN architectures. Moreover, GCN achieves the best results on most datasets, which explains our choice of GCN as the backbone.
>
> |        | Coauthor CS | Amazon Photo | Cora | Pubmed | DBLP | Citeseer |
> | ---- | ---- | ---- | ---- | ---- | ---- | ---- |
> | GCN | **87.4 ± 0.8** | 87.3 ± 0.5 | **80.9 ± 0.8**  | **80.3 ± 0.5**  | 80.1 ± 0.6  | **71.5 ± 0.5**  |
> | GAT | 86.8 ± 1.2 | **87.4 ± 0.8**  |80.2 ± 1.1  | 80.1 ± 0.7  |79.8 ± 0.7  | 71.0 ± 0.7  |
> | GIN | 87.1 ± 0.7 | 87.0 ± 0.6  |80.6 ± 0.9  | 79.9 ± 0.5  |**80.3 ± 0.5**  | 71.3 ± 0.4  |
>
> > **Q4**: Some spelling errors in individual words need to be checked.
>
> **R**: Thank you for the comment. We have done our best to correct the typos.

---

> > ### Author Rebuttal · Reviewer_ynSy · 2026-04-02
> >
> > The author's detailed explanation further addressed my concerns, and I am very satisfied with it, so I have raised my score accordingly.

---

> > > ### Author Response · Authors · 2026-04-05
> > >
> > > Thanks for your feedback and increasing the rating! We will properly include all the rebuttal contents in the revised version, following your valuable suggestions.

---

### Official Review · Reviewer_wz2v · 2026-03-07

**Soundness:** 3
**Presentation:** 4
**Significance:** 3
**Originality:** 4
**Overall Recommendation:** 5
**Confidence:** 5

**Summary:**

This paper introduces ICGNN, a robust graph neural network designed to handle noisy and limited labels in graph-structured data. The method detects label noise by measuring an influence contradiction score (ICS) from both structural and attribute-level graph diffusion, and uses a GMM to identify noisy nodes. A soft label correction strategy aggregates neighbor predictions to adjust detected noisy labels, while pseudo-labeling on unlabeled nodes provides additional supervision. Extensive experiments on benchmark datasets demonstrate that ICGNN outperforms existing methods under various noise and label rates.

**Compliance With Llm Reviewing Policy:**

Affirmed.

**Final Justification:**

The authors’ response has addressed my concerns, and I will raise my score.

**Key Questions For Authors:**

See Weaknesses.

**Limitations:**

Yes

**Strengths And Weaknesses:**

Strengths:
1. The proposed ICS leverages both graph structure and node attributes to quantify label credibility, enabling more accurate identification of noisy nodes than traditional loss-based methods.
2. ICGNN uses a confidence-weighted combination of original labels and neighbor predictions, reducing the risk of confirmation bias.
3. By generating pseudo-labels for unlabeled nodes, the model gains additional supervision, improving performance in semi-supervised settings with scarce clean labels.
4. ICGNN consistently outperforms competitive baselines across multiple datasets and noise settings, with ablation studies confirming the contribution of each component.

Weaknesses:
1. The ICS criterion proposed in Eq. 3 uses the category information containing noise. Will this interfere with the detection results?
2. In the GMM fitting, why is the component 2 and what is the justification for doing so?
3. Does the utilization of neighbor information further introduce and diffuse noise when performing label correction?
4. Does the EM algorithm outside the gradient increase the complexity of the algorithm?

---

> ### Author Rebuttal · Authors · 2026-03-31
>
> We appreciate your thoughtful comments and suggestions. We will make every effort to address your concerns as follows.
>
> > **Q1**: The ICS criterion proposed in Eq. 3 uses the category information containing noise. Will this interfere with the detection results?
>
> **R**: Thanks for the valuable feedback! Noisy labels do not interfere with the detection results; on the contrary, their presence motivates us to establish a noise detection criterion guided by noisy labels. Specifically, the proposed ICS is theoretically grounded in the homophily principle and global influence patterns inherent in graph-structured data. Under homophily, correctly labeled nodes are expected to receive influence primarily from nodes of the same class, whereas noisy labels disrupt this consistency by introducing substantial cross-class influence. ICS quantifies this inconsistency by leveraging both structure-level and attribute-level diffusion matrices, which capture multi-hop influence propagation. Consequently, a higher ICS value indicates a greater contradiction between structural and attribute similarities and the node's annotated label, suggesting a higher possibility of the node having a noisy label.
>
> > **Q2**: In the GMM fitting, why is the component 2 and what is the justification for doing so?
>
> **R**: Thanks for the question! In our proposed method, we identify noisy labels based on the computed ICS criterion values. Since different datasets require different division thresholds, we leverage GMM to perform learnable soft-threshold clustering to avoid manually-set hard thresholds. The number of components is set to 2 because our objective is to classify nodes into clean and noisy categories, and output the probabilities of nodes being classified as clean or noisy, thereby facilitating more robust noise label correction in subsequent steps.
>
> > **Q3**: Does the utilization of neighbor information further introduce and diffuse noise when performing label correction?
>
> **R**: Thanks for the valuable question! Our method is not prone to introducing and diffusing noise. In Eq. (7), we use the predicted class assignments of neighboring nodes rather than their true labels. It helps to avoid introducing noisy labels during the label correction process to some extent. By utilizing the predicted assignment information, we also enhance the utilization of information from unlabeled nodes.
>
> This strategy is further supported by the principle that nodes with stronger connectivity or higher similarity are more likely to share the same label. Therefore, using predicted assignments from neighboring nodes provides a reliable and robust basis for correction. In this sense, the process is essentially self-supervised and self-enhancing: the model iteratively refines its own predictions to guide supervision. As a result, effective optimization is achieved while mitigating the negative impact of noisy labels.
>
> > **Q4**: Does the EM algorithm outside the gradient increase the complexity of the algorithm?
>
> **R**: Thanks for the question. The EM algorithm does not introduce additional complexity; it only takes a very small amount of runtime. As described in Section 4.7, the number of iterations in the EM algorithm is typically less than 10 to achieve satisfactory results, so its complexity is negligible compared to the overall complexity. Furthermore, we also visualize in Figure 6 the comparison of runtime and memory consumption between our method and competitive baselines. Our method achieves higher computational efficiency and comparable memory consumption while maintaining excellent performance.

---

> > ### Author Rebuttal · Reviewer_wz2v · 2026-04-02
> >
> > The authors’ response has addressed my concerns, and I will raise my score.

---

> > > ### Author Response · Authors · 2026-04-05
> > >
> > > Thanks for your feedback and increasing the rating! We will properly include all the rebuttal contents in the revised version, following your valuable suggestions.

---

### Official Review · Reviewer_v2n7 · 2026-03-10

**Soundness:** 3
**Presentation:** 3
**Significance:** 3
**Originality:** 3
**Overall Recommendation:** 5
**Confidence:** 4

**Summary:**

This paper addresses the problem of learning Graph Neural Networks (GNNs) in the presence of noisy labels, a critical challenge for real-world graph data. The authors propose a method named ICGNN, which introduces a novel "Influence Contradiction Score" (ICS) to detect potentially mislabeled nodes. This score leverages a graph diffusion matrix to quantify the consistency of a node's influence on its neighbors. A two-component Gaussian Mixture Model (GMM) is then used to classify nodes as clean or noisy based on their ICS values. For nodes identified as noisy, a soft correction strategy is applied, combining the original noisy label with an aggregated prediction from neighboring nodes. The method also incorporates pseudo-labeling for unlabeled nodes to provide additional supervision. Extensive experiments on six benchmark datasets under both uniform and pair noise settings demonstrate that ICGNN achieves state-of-the-art or competitive performance compared to existing methods.

**Compliance With Llm Reviewing Policy:**

Affirmed.

**Key Questions For Authors:**

Pelease refer to the weakness.

**Limitations:**

The primary limitations are the method's increased conceptual and computational complexity, the potential for confirmation bias in its correction strategy, and a reliance on the GMM's bimodal assumption that may not always hold. The paper's empirical validation is strong but focused on homophilous citation and product networks; its performance on heterophilous graphs, where neighbor influence is less indicative of label correctness, remains an open question. The scalability to graphs with a very large number of labeled nodes is also a potential concern.

**Strengths And Weaknesses:**

Strengths:

(1) Well-Motivated Detection Mechanism: The core idea of using an "Influence Contradiction Score" is a novel contribution to the graph noisy label literature. By quantifying the disagreement between a node's label and its influence on neighbors at both structural and feature levels, the method provides a principled way to identify noisy nodes that goes beyond simple prediction confidence or small-loss heuristics. The use of a graph diffusion matrix to capture global influence is a thoughtful design choice.

(2) Comprehensive and Rigorous Experimental Evaluation: The experimental setup is a clear strength. The authors evaluate their method on six diverse datasets (citation, co-author, co-purchase) under two different noise types (uniform and pair). The baseline comparison is extensive, including 9 main baselines and an additional 12 in the appendix, covering classic GNNs, general noisy-label methods, and recent graph-specific approaches.

(3) Architectural Agnosticism: The paper correctly notes that ICGNN is a plug-and-play module that can be applied to any GNN backbone, as demonstrated in Appendix D.1. This is a significant practical advantage, enhancing the method's applicability and potential impact.

(4) Strong Empirical Performance: ICGNN consistently outperforms all baselines across almost all datasets and settings. The gains, while sometimes modest, are statistically significant in most cases, validating the practical utility of the proposed approach.


Weaknesses:
(1) Potential Circular Dependency in "Soft Correction": The soft correction strategy in Eq. (6) uses the model's predictions from the current epoch to correct the labels for the same epoch's loss calculation (Eq. 8). This creates a potential for confirmation bias, where the model's own (possibly incorrect) early predictions are used to reinforce itself. A more robust approach,  is to use predictions from a previous epoch (a "teacher" model) or an Exponential Moving Average (EMA) of predictions to correct labels for the current epoch, thereby breaking the circular dependency.

(2) Limited Discussion on the Choice of GMM: The paper defaults to a two-component GMM, implicitly assuming a clear separation between clean and noisy nodes based on ICS. However, the ICS distribution may not always be bimodal. In cases of high noise or class ambiguity, the distribution could be more complex. The paper would be strengthened by an analysis of the fitted GMM components (e.g., visualizing the means μ_q and showing that the component with the lower mean consistently corresponds to clean nodes) across different datasets and noise levels. This would validate the fundamental assumption of the detection module.

(3) Scalability Concerns: The complexity analysis states an O(EN + L^3) term due to the node-wise optimization for α_i. While the authors claim L ≪ N, this may not hold for all semi-supervised scenarios. On large graphs with tens of thousands of labeled nodes (e.g., in active learning or fully-supervised settings), an O(L^3) term could become a severe bottleneck. The paper should discuss the scalability of this optimization step and its practical limits.

---

> ### Author Rebuttal · Authors · 2026-03-31
>
> > **Q1**: A more robust approach is to use previous epoch predictions or EMA to correct labels, breaking circular dependency.
>
> **R**: Thanks for the valuable suggestion. We conducted careful experiments to evaluate this suggestion. Specifically, we replaced $h^{(t)}(\mathbf z_i)$ in Eq. (6) with (1) the previous epoch prediction: $h^{(t-1)}(\mathbf z_i)$, and (2) EMA: $\epsilon h^{(t-1)}(\mathbf z_i) + (1-\epsilon)h^{(t)}(\mathbf z_i)$, and performed comparative experiments on Cora, Pubmed, and Citeseer. Our strategy shows no significant disadvantage compared to the two variants in the table below, demonstrating the robustness of our confidence-driven cleaning. This is because ICS and confidence $\hat{\beta}_i$ are independent of prediction, controlling the cleaning strength. Even if the prediction contains errors, the detection can still provide independent constraints to prevent confirmation bias and error accumulation. On the contrary, introducing the two alternative strategies may lead to information lag and hyperparameter tuning issues associated with the momentum coefficient. We will add this discussion in the future version.
>
> ||Uniform|||Pair|||
> |-|-|-|-|-|-|-|
> | Dataset | previous epoch | EMA | Ours|previous epoch|EMA | Ours|
> | Cora | 79.8±0.9 | 80.3±0.7 | 80.9±0.8| 79.1±1.0 | 78.8±0.9 |79.4±0.7|
> | Pubmed | 80.2±0.6 | 80.1±0.6 |80.3±0.5| 79.4±0.3 | 79.3±0.5 |80.6±0.3|
> | Citeseer | 71.0±0.8 | 71.2±0.8 |71.5±0.5| 70.9±0.5 | 70.7±0.6 |70.7±0.7|
>
> > **Q2**: The paper would be strengthened by validating that the lower-mean component corresponds to clean nodes across datasets and noise levels.
>
> **R**: Thanks for the valuable suggestion. As demonstrated in Sec. 4.6, we validated our noise detection method by visualizing the relationship between noisy/clean nodes with loss (small-loss criterion) and our ICS-based confidence. The confidence refers to the probability that a node is clean, obtained via GMM fitting, which is positively correlated with the node's similarity to the lower mean. Fig. 5 shows on Amazon Photo (40% noise) that clean and noisy samples form well-separated confidence clusters, enabling clear differentiation. Additional visualizations on DBLP and Citeseer under 20%, 30%, and 40% noise rates (see https://anonymous.4open.science/r/icgnn-rebuttal/icgnn_rebuttal.pdf) further confirm that our method consistently separates clean from noisy nodes across different datasets and noise levels. We will incorporate these additional results into the future version.
>
> > **Q3**: Scalability and practical limits should be discussed.
>
> **R**: Thanks for your valuable comment! The issue raised by the reviewer is indeed a critical one, which reveals a limitation of our method in scenarios such as fully-supervised settings. The $O(L^3)$ term mainly originates from the computation of the diffusion matrix $\mathbf R$ in Eq. (4). To reduce the complexity, we can first use low-precision methods (such as approximating the diffusion matrix via power iteration or Monte Carlo methods, and reducing the number of iterations in power iteration or the number of walks in Monte Carlo methods) to quickly screen candidate noisy nodes, and then perform high-precision computation on the candidate set. Additionally, high-confidence nodes can reuse historical ICS values, updating only low-confidence nodes, which further reduces the computational cost in subsequent epochs. Moreover, we can directly sparsify the adjacency matrix through sampling or truncation, thereby reducing the complexity of diffusion matrix computation to linear [1]. We will incorporate these discussions into the future version.
>
> > **Q4**: The method's performance on heterophilous graphs remains an open question.
>
> **R**: Thanks for your comment. Our method does not simply rely on the homophily assumption; instead, it identifies label noise by leveraging contradictions in both structural and attribute information. In heterophilous graphs, attribute features often align with true class even when structure does not. Thus, our ICS can alleviate interference from heterophilous graph structures: a large ICS value arises only when both structural and attribute contradictions are substantial. Even when ICS is affected by heterophily, our GMM-guided soft-threshold mechanism avoids discarding correct information, mitigating misclassification impact. Empirically, we show the performance comparison between our ICGNN and competitive baselines on the heterophilous graph Cornell in Table 5, where ICGNN demonstrates good applicability across different types of graphs.
>
> We acknowledge the method is not designed for heterophilous graphs and only partially alleviates homophily reliance. Future work will develop dedicated detection criteria for noisy heterophilous graphs, and we will incorporate this discussion into the revision.
>
> [1] Johannes Klicpera, Stefan Weißenberger, and Stephan Gunnemann. Diffusion improves graph learning.

---

> > ### Author Rebuttal · Reviewer_v2n7 · 2026-04-05
> >
> > The authors’ response has addressed my concerns, and I will remain my positive score.

---

> > > ### Author Response · Authors · 2026-04-05
> > >
> > > Thanks for your positive feedback! We will properly include all the rebuttal contents in the revised version, following your valuable suggestions.

---

### Official Review · Reviewer_YtDL · 2026-03-12

**Soundness:** 3
**Presentation:** 3
**Significance:** 2
**Originality:** 2
**Overall Recommendation:** 4
**Confidence:** 4

**Summary:**

The paper proposes ICGNN, a framework for semi-supervised node classification with noisy and scarce labels. It uses an influence contradiction score (ICS) with GMM-based confidence estimation to detect noisy labels, then combines original labels and neighbor-based predictions for robust training; experiments show consistent gains across multiple benchmarks and noise settings.

**Compliance With Llm Reviewing Policy:**

Affirmed.

**Final Justification:**

I thank the authors for their detailed rebuttal and the additional experiments provided.  I am satisfied with the responses and am happy to raise my score.

**Key Questions For Authors:**

Please see the weaknesses section above.

**Limitations:**

No. The paper provides no discussion of limitations.

**Strengths And Weaknesses:**

Strengths:
1) The paper is generally clearly written, and both the methodological pipeline and the experimental setup are easy to follow.

2) The implementation details and hyperparameter settings are described in sufficient detail, which supports reproducibility.

Weaknesses：
1) The novelty is limited. The method mostly combines existing components(PPR-based structural propagation, representation-space KNN neighborhoods,and GMM-based noise separation) and it is still unclear what the main new technical contribution is.

2) The motivation is not fully convincing. The paper does not clearly explain why the proposed contradiction score can distinguish mislabeled nodes from hard or unusual nodes, or why this combination is better than simpler alternatives.

3) There is a mismatch between the motivation and the experiments. The method is motivated by graph-structural contradiction, but the experiments only use standard synthetic noise such as uniform and pair noise, which do not really match the claimed setting.

4) The method seems to rely heavily on homophily. On heterophilous graphs, strong influence from other classes may simply come from the graph structure itself, not from label noise. The paper does not provide enough analysis for this case.

5) The theoretical support is weak. It is still unclear under what graph or noise settings the proposed ICS can really separate clean from noisy nodes. The paper mainly relies on intuition and empirical results.

---

> ### Author Rebuttal · Authors · 2026-03-31
>
> > **Q1**: Its novelty is unclear.
>
> **R**: Thanks for the comment. While PPR, KNN, and GMM are existing techniques, the novelty lies in how they are synergistically integrated into a unified framework for noise detection and cleaning, rather than being a simple stacking of off-the-shelf tools. As described in Lines 71-95, existing methods implicitly use structural or attribute information to mitigate noise but cannot achieve explicit detection and multi-source utilization simultaneously. In contrast, our proposed ICS, jointly models the structure and attribute diffusion matrices, forming a dual-contradiction detection criterion.
>
> Moreover, existing methods lack robust label cleaning and suffer from detection-cleaning disconnect. Our method integrates detection into cleaning via confidence scores, enabling dynamic adjustment and collaborative optimization. This end-to-end closed-loop design is our core contribution.
>
> > **Q2**: Why the contradiction score distinguishes mislabeled nodes from hard/unusual nodes, and why it is better than simpler alternatives?
>
> **R**: Thanks for the insightful comment! First, our proposed ICS detects label noise from the perspective of influence contradiction at both the structural and attribute levels. Hard or unusual nodes typically show contradiction in only one dimension: hard nodes have ambiguous attributes but consistent neighbors; heterophilous nodes have cross-class connections but consistent attributes. Therefore, ICS is generally able to distinguish mislabeled nodes from hard or unusual nodes, because a node is detected as noisy by ICS only when both structural and attribute contradictions are strong.
>
> Furthermore, Sec. 4.9 provided an experimental analysis of ICS's effectiveness in identifying noisy samples. Fig. 7(a) shows lower ICS nodes lie farther from class boundaries, indicating our method effectively isolates clean nodes (as boundary nodes are more prone to noise). Fig. 7(b) shows loss fails to provide clear separation, as high-loss nodes do not consistently align with class boundaries. ICS provides a nuanced view of node consistency that goes beyond what the loss metric can capture.
>
> > **Q3**: Experiments only use common noise.
>
> **R**: Thanks for the comment! Our noise detection is derived from the dual contradictions at both structural and attribute levels, which originates from the inherent properties of graph structures themselves. We do not claim that it is designed to handle any specific type of noise. Therefore, we used standard uniform and pair noises to demonstrate our method's superiority, as validating under general noise settings better reflects its effectiveness.
>
> > **Q4**: On heterophilous graphs, influence from other classes may stem from structure, not noise. Analysis is insufficient.
>
> **R**: Thanks for your comment. Our method does not simply rely on the homophily assumption; instead, it identifies label noise by leveraging contradictions in both structural and attribute information. In heterophilous graphs, attribute features often align with true class even when structure does not. Thus, our ICS can alleviate interference from heterophilous graph structures: a large ICS value arises only when both structural and attribute contradictions are substantial. Even when ICS is affected by heterophily, our GMM-guided soft-threshold mechanism avoids discarding correct information, mitigating misclassification impact. Empirically, we show the performance comparison between our ICGNN and competitive baselines on the heterophilous graph Cornell in Table 5, where ICGNN demonstrates good applicability across different types of graphs.
>
> We acknowledge the method is not designed for heterophilous graphs and only partially alleviates homophily reliance. Future work will develop dedicated detection criteria for noisy heterophilous graphs, and we will incorporate this discussion into the revision.
>
> > **Q5**: It is unclear when ICS separates clean from noisy nodes.
>
> **R**: Thanks for the comment. Assume that there exists a constant $\delta \in (0,1]$ such that for any node $i$, $\frac{1}{|C_ {y_ i^* } |} \sum_ {k \in C_ {y_ i^* }}\alpha\mathbf{T}_ {ki}+(1-\alpha)\mathbf{R}_ {ki}\geq\delta$, corresponding to a joint structural- and attribute-level homophily assumption, where $y_i^*$ is the true label. It can be theoretically shown that for clean nodes, ICS$_i \leq 1-\delta$, while for noisy nodes, the ICS$_i > \delta$. Therefore, when $\delta$ is relatively large, the ICS values of clean and noisy nodes lie in two disjoint intervals, enabling direct separation by setting a threshold. When $\delta$ is relatively small, the expected values of the two classes still differ, and the distribution of ICS can be effectively modeled by a GMM to achieve soft-threshold detection. This theoretical basis will be supplemented with a proof in the future version.

---

> > ### Author Rebuttal · Reviewer_YtDL · 2026-04-03
> >
> > I thank the authors for sketching out the theoretical argument in your rebuttal, the idea that clean and noisy nodes lie in disjoint ICS intervals when $\delta$ is bounded away from zero is a helpful framing. That said, could you say a bit more about when we'd actually expect this separation to hold in practice? Specifically, what conditions on the graph structure or noise rate are needed to keep $\delta$ from collapsing toward zero? Even an informal discussion of this would go a long way toward helping readers trust that the step is reliably detecting noise rather than just fitting whatever distribution happens to show up.

---

> > > ### Author Response · Authors · 2026-04-04
> > >
> > > > **Q**: what conditions on the graph structure or noise rate are needed to keep $\delta$ from collapsing toward zero?
> > >
> > > **R**: Thanks for the constructive comment. Under what conditions $\delta$ can stay away from zero is indeed a question worth discussing. In fact, if (1) the distribution of edges in the graph exhibits a certain class tendency, e.g., most neighbors of node $i$ share the same true label as node $i$, or (2) nodes of the same class tend to cluster in the representation space while nodes of different classes are well separated, then $\mathcal{A} = \frac{1}{|C_ {y_ i^* }|} \sum_ {k \in C_ {y_ i^* }} ( \alpha \mathbf{T}_ {ki} + (1-\alpha) \mathbf{R}_ {ki} )$ will be relatively large, and consequently $\delta$ will stay away from zero. The influence of label noise on $\delta$ only comes from the supervision of representation learning during training. In our training process, we use corrected labels for supervision, accompanied by high-confidence pseudo-labels. Therefore, the negative impact of noise can be gradually mitigated, and the possible value of $\delta$ will increase during training. When the noise rate is not too high, this strategy leads to more discriminative ICS intervals and achieves satisfactory noise detection performance.
> > >
> > > To validate this, on one hand, we computed the values of $\mathcal{A}$ on Cora (pair noise) and Citeseer (uniform noise) under 20%, 40%, and 60% noise rates. The minimum values obtained are 0.63, 0.53, 0.36 (Cora) and 0.56, 0.48, 0.27 (Citeseer); the averaged values are 0.91, 0.86, 0.75 (Cora) and 0.87, 0.79, 0.68 (Citeseer). On the other hand, as shown in Fig. 3, Fig. 8, and Tab. 7, when the noise rate is no more than 40%, our ICGNN maintains reasonably small performance degradation, indicating that a relatively large $\delta$ is preserved and ICS can well separate clean nodes from noisy ones. When the noise rate reaches 60% or higher, the performance degradation becomes more pronounced, and the discriminative power of ICS weakens. Nevertheless, compared to competitive baselines, our ICGNN still achieves superior performance, demonstrating the robustness and superiority of our GMM-guided cleaning strategy.
> > >
> > > We will also incorporate these discussions into the future version.

---

### Decision · Program_Chairs · 2026-04-30

**Decision:**

Accept (regular)

**Comment:**

The paper proposes ICGNN, a framework for semi-supervised node classification with noisy and scarce labels. It uses an influence contradiction score (ICS) with GMM-based confidence estimation to detect noisy labels, then combines original labels and neighbor-based predictions for robust training; experiments show consistent gains across multiple benchmarks and noise settings. All the reviewers hold positive comments on it. I would recommend it as Accepted.